# REWARD-AGNOSTIC PROMPT OPTIMIZATION FOR TEXT-TO-IMAGE DIFFUSION MODELS

## ABSTRACT

We investigate a general approach for improving user prompts in text-to-image (T2I) diffusion models by finding prompts that maximize a reward function specified at test-time. Although diverse reward models are used for evaluating image generation, existing automated prompt engineering methods typically target specific reward configurations. Consequently, these specialized designs exhibit suboptimal performance when applied to new prompt engineering scenarios involving different reward models. To address this limitation, we introduce RATTPO (Reward-Agnostic Test-Time Prompt Optimization), a flexible test-time optimization method applicable across various reward scenarios without modification. RATTPO iteratively searches for optimized prompts by querying large language models (LLMs) *without* requiring reward-specific task descriptions. Instead, it uses the optimization trajectory and a novel reward-aware feedback signal (termed a "hint") as context. Empirical results demonstrate the versatility of RATTPO, effectively enhancing user prompts across diverse reward setups that assess various generation aspects, such as aesthetics, general human preference, or spatial relationships between objects. RATTPO surpasses other test-time search baselines in search efficiency, running 4.8 times faster than naive reward-agnostic test-time search baseline on average. Furthermore, with sufficient inference budget, it can achieve comparable performance to learning-based baselines that require reward-specific fine-tuning.

## 1 INTRODUCTION

Recent advancements in text-to-image (T2I) diffusion models have enabled the generation of high-quality and diverse images from user prompts (Rombach et al., 2022; Ramesh et al., 2022; Podell et al., 2024; Black Forest Labs, 2024). Despite the success, their output is heavily reliant on the input prompts (Liu & Chilton, 2022; Oppenlaender, 2024; Diab et al., 2022), often exhibiting noticeable fluctuations in generation quality in response to subtle changes in prompts (Mahajan et al., 2024; Cao et al., 2023) that do not alter the underlying semantics. To improve the generation process to follow user intent and yield desired visual outputs, one needs to iteratively improve the initial prompt through a trial-and-error loop, a process commonly referred to as prompt engineering.

Prompt engineering refines prompts to guide a T2I model to produce *better* images. The quality of images is often measured by diverse reward models evaluating aspects like human preference (Xu et al., 2023; Schuhmann, 2022; Kirstain et al., 2023; Wu et al., 2023), text-to-image alignment (Radford et al., 2021; Huang et al., 2025; Hu et al., 2023; Cho et al., 2024), and their implicit or explicit combinations (Hao et al., 2023; Ma et al., 2025; Huang et al., 2025; Sun et al., 2024; Chen et al., 2023). Based on these reward models, several works (Yun et al., 2025; Hao et al., 2023; Mo et al., 2024; Mañas et al., 2024; Wang et al., 2024) have proposed automated prompt engineers that rewrite user prompts into enhanced versions capable of generating high-reward images.

However, many of the aforementioned automated prompt engineering techniques (Yun et al., 2025; Hao et al., 2023; Mañas et al., 2024; Mo et al., 2024) are tailored for particular reward models. Consequently, these specialized methods often exhibit suboptimal performance when applied to new scenarios involving different reward functions, which limits their general applicability. This limitation highlights the need for a versatile prompt optimization technique that can adapt to diverse reward functions at test-time without requiring retraining or manual, reward-specific adjustments.

To address this challenge, we introduce Reward-Agnostic Test-Time Prompt Optimization (RATTPO), a flexible test-time optimization approach for automated prompt engineering under diverse reward functions that are only specified at test time. The method iteratively refines an initial user prompt by querying large language models (LLMs), without relying on explicit reward-specific task descriptions that are often human-crafted for better guiding the LLMs. Instead, it conditions the LLM-based optimizer on the historical trajectory of previously attempted prompts and their corresponding reward scores, along with a novel reward-aware feedback signal we term a "hint." Each hint is a concise textual strategy for increasing the reward, analogous to a manually written task description, but generated on-the-fly by an independent LLM during optimization. This design removes the need for manual rewriting while still providing reward-aware guidance to the optimizer.

Our contributions are threefold: First, we propose RATTPO, a training-free and gradient-free automated prompt engineer that is readily applicable to diverse reward setups without requiring reward-specific adjustments or training. Second, we introduce "hint", a novel, reward-aware self-feedback mechanism to guide the prompt engineering process. The hint offsets the absence of a reward-specific design by estimating strategy for improving reward from optimization trajectory. Third, we empirically demonstrate RATTPO's effectiveness and versatility across various reward settings, including human preference, text-to-image consistency, and holistic evaluation using a multimodal LLM. Our extensive experiments demonstrate that RATTPO can effectively improve the initial prompt with respect to diverse rewards, and shows higher search efficiency compared to other test-time search baselines.

## 2 RELATED WORK

**LLM-Based Optimization**    Recent works have leverage the strong instruction-following (Ouyang et al., 2022; Sanh et al., 2022; Qin et al., 2024) and in-context learning capabilities (Brown et al., 2020; Wei et al., 2022; Dong et al., 2022) of LLMs for various optimization tasks (Yang et al., 2024; Du et al., 2024; Mañas et al., 2024; He et al., 2024; Zhang et al., 2023; Liu et al., 2023). Among these, OPRO (Yang et al., 2024) is closely related to our approach, proposing an optimization-by-prompting framework that iteratively queries LLMs using previous optimization history as context. RATTPO extends this concept by constructing a dual-LLM optimization loop with a novel feedback mechanism, tackling the unique problem of building a reward-agnostic prompt engineer for T2I models.

**Automated Prompt Engineering for Diffusion Models**    The output generated by diffusion models often deviates from user intention or preference, as captured by diverse reward models assessing human preference (Xu et al., 2023; Schuhmann, 2022; Kirstain et al., 2023; Wu et al., 2023), text-to-image consistency (Radford et al., 2021; Huang et al., 2025; Hu et al., 2023; Cho et al., 2024), and other criteria (Hao et al., 2023; Ma et al., 2025; Huang et al., 2025; Sun et al., 2024; Chen et al., 2023). To bridge this gap without manual prompting, several studies have focused on automated prompt engineering for a given reward, employing either training-based or test-time approaches.

Learning-based methods like Promptist (Hao et al., 2023) define a heuristic reward model and train a language model via reinforcement learning (RL). Similarly, Mo et al. (2024) propose finetuning a language model using RL to utilize specialized prompt formats, while PAG (Yun et al., 2025) focuses on the diversity of resulting prompts. While effective for trained rewards, these methods incur considerable training costs (*e.g.*, approximately 4 GPU days in Yun et al. (2025)). We also empirically observe that their transferability to novel reward scenarios is limited (see Sec. 4.2).

Instead of training language models, a few works seek alternatives that utilize test-time computation. DPO-Diff (Wang et al., 2024) constrains a search space with LLM-generated synonyms or antonyms and optimizes the negative prompt via gradient descent employing several optimization tricks. While this approach makes the search tractable, its performance is inherently limited as the reduced search space may exclude optimal prompt candidates. OPT2I (Mañas et al., 2024) is similar to our approach in that it also employs an LLM to optimize user prompts at test time. Specifically, OPT2I constructs an iterative loop in which an LLM refines the initial prompt based on history from previous iterations to enhance text-to-image consistency. Despite being a test-time approach, the LLM in OPT2I is tied with human-crafted query prompts and thus they are tailored for and evaluated under two specific choices of reward functions, leaving its applicability to unseen, diverse rewards unanswered.

---

**Algorithm 1** RATTPO: Reward-Agnostic Test-Time Prompt Optimization

---

1: **Input:** User prompt $p_0$, T2I model $G$, reward model $R$, optimization iterations $N$
2: **Initialize:** history $\leftarrow \emptyset$, hint $\leftarrow$ null
3: **for** $t = 1$ **to** $N$ **do**
4:      $\text{context}_o \leftarrow$ SampleTrajectories(history)
5:      candidate_prompts $\leftarrow \mathcal{L}_o(p_0, \text{context}_o, \text{hint})$      $\triangleright$ Optimizer LLM proposes prompts
6:      **for** each $p$ in candidate_prompts **do**
7:          $\text{score} = \mathbb{E}_{I \sim G(p)} R(I, p_0)$      $\triangleright$ Generate images and compute score
8:          history $\leftarrow$ history $\cup \{p, \text{score}\}$      $\triangleright$ Update history
9:      **end for**
10:     hint $\leftarrow \mathcal{L}_h($SampleTrajectories(history)$)$      $\triangleright$ Hint-generator LLM generates hint
11: **end for**
12: **Return:** Best prompt $\hat{p}$      $\triangleright$ Select the best-scoring prompt from history

---

## 3 METHOD

### 3.1 PROBLEM SETUP

We consider the problem of improving user prompts for text-to-image (T2I) generative models. Given an initial prompt $p_0$, a T2I generative model $G$, and a reward function $R$, prompt engineering aims to produce an enhanced prompt $\hat{p}$ that maximizes the expected reward of the generated images while preserving the semantics of the original prompt. Formally, this can be cast as the following optimization problem over the set of prompts that preserve the semantics $\mathcal{S}(p_0)$:

$$\hat{p} = \arg\max_{p \in \mathcal{S}(p_0)} \mathbb{E}_{I \sim G(p)}[R(I, p_0)]. \tag{1}$$

Automating the prompt engineering process has often been considered under predefined reward setups. A prominent example is finetuning a language model with respect to reward $R$ during training (Hao et al., 2023; Mo et al., 2024; Yun et al., 2025), resulting in a dedicated prompt engineer for the specific reward and T2I model it was trained on. While it is technically possible to use them for different reward models at test-time without reward-specific re-training, they show suboptimal performance when there is a large shift in the target reward. On the other hand, test-time approaches offer greater flexibility by adapting to new rewards on-the-fly, but previous methods either limit the search space for tractability (Wang et al., 2024), which weakens performance, or are tailored for specific reward choices (Mañas et al., 2024), necessitating manual configuration for application to a new reward.

In contrast, we focus on the more challenging scenario of building a *reward-agnostic* automated prompt engineer that is capable of solving Eq. 1 for a broad range of unknown reward functions $R$, so it can be directly applied to diverse prompt engineering scenarios without any further modification. Our aim is to handle real-world application scenarios, where reward models with different evaluation rubrics, potentially personalized ones, are continually developed and deployed for capturing fine-grained preferences. In such application scenarios, a principled reward-agnostic prompt engineer can be trained or designed once and later used for downstream reward models in general, avoiding the need to train separate dedicated prompt engineers again.

### 3.2 BUILDING REWARD-AGNOSTIC PROMPT OPTIMIZER

To build a reward-agnostic yet effective automated prompt engineer, we choose a test-time search approach equipped with LLMs for efficient exploration of the large search space $\mathcal{S}(p_0)$. Our optimization iteration consists of two LLMs, one for proposing prompt candidates and another for providing reward-specific feedback. At each iteration, the *optimizer* LLM $\mathcal{L}_o$ first proposes a set of promising prompt candidates that are used to generate images. The generated images are then evaluated using reward model $R$ to score the proposed prompts. The *hint-generator* LLM $\mathcal{L}_h$ is used to provide feedback to the optimizer LLM by describing the reward function based on optimization history. Below, we describe each component and its design choices (the complete algorithm is presented in Alg. 1; prompt templates for querying LLMs can be found in App. B).

**The Optimizer LLM** Motivated by the successful application of LLMs to various optimization problems (Yang et al., 2024; Du et al., 2024; Mañas et al., 2024; He et al., 2024; Zhang et al., 2023;

Liu et al., 2023), we exploit the strong in-context learning (ICL) capabilities of LLMs (Brown et al., 2020; Wei et al., 2022; Dong et al., 2022) to solve Eq. 1. Specifically, we query the optimizer LLM to enhance the initial prompt in several distinct ways by using top-$k$ history from previous iterations and feedback from the hint-generator LLM. By prompting the optimizer LLM with a handful of optimization history examples, we bias its output prompt candidates towards more promising regions of the search space. This in-context policy shift uses examples generated on-the-fly and therefore does not require reward-specific finetuning, aligning with our goal of building a reward-agnostic automated prompt engineer. Furthermore, the optimization process is both training-free and gradient-free, making it applicable to diverse reward models that are often non-differentiable (*e.g.*, visual question answering models) or only available via forward API call (*e.g.*, proprietary vision-language models).

**Guiding the Optimizer LLM with the Hint-Generator LLM**   The optimizer LLM is controlled by a *meta-prompt*, an instruction used for querying it (Yang et al., 2024). Previous work (Yang et al., 2024; Fernando et al., 2023; Du et al., 2024; He et al., 2024; Mañas et al., 2024) has typically relied on predefined task descriptions and objectives to instruct the optimizer LLM (Yang et al., 2024). While removing such descriptions is necessary for our purposes, it can also reduce optimization performance. Therefore, we keep the task description general and concise, and instead augment it with another type of reward-specific optimization signal.

Inspired by successful self-feedback techniques in NLP literature (Madaan et al., 2023; Wang et al., 2023a; Shinn et al., 2023), we introduce the hint-generator LLM to compensate for the potential loss in search efficiency. To be specific, we ask hint-generator LLM to response with how we can improve the score, given the context of optimization history (see Fig. 4 for example). When constructing the context, we stick to a simple design of also using the search history obtained from the optimizer LLM, since it already contains information about the optimization objective. To help the hint-generator LLM better identify the reward function, we provide random subset of histories as context that include both good and bad examples. Besides the empirical gain in search efficiency, the hint is formatted as natural language feedback and is therefore human-interpretable. This transparency is often beneficial, since human prompt engineers can review the generated hints and later use them for manual prompt engineering.

# 4 EXPERIMENT

## 4.1 EXPERIMENTAL SETTING

**Reward Models**   To evaluate the effectiveness of RATTPO across diverse scenarios, we consider various reward models as optimization targets, primarily categorized into human preference, text-to-image consistency, and holistic assessment using multimodal LLMs (MLLMs).

- *Human Preference*: We consider the Promptist Reward (Hao et al., 2023) and ImageReward (Xu et al., 2023). Promptist Reward combines Aesthetic Score (Schuhmann, 2022) and CLIP Score (Radford et al., 2021) to balance image aesthetics with faithfulness to the prompt. ImageReward is trained on a human-annotated preference dataset to capture general human preferences.

- *Text-to-Image Consistency*: We consider reward models with different scoring mechanisms. First, DSG (Cho et al., 2024) generates atomic questions with a dependency graph from a given prompt, and then utilizes a visual question answering model for scoring. Secondly, we adopt three scorers from T2I-CompBench++ (Huang et al., 2025) to assess 2D/3D spatial relationships and numeracy (UniDet2D, UniDet3D, and UniDetNumeracy, respectively) using an object detection model.

- *Holistic MLLM Assessment*: We assess image quality using LLMGrader, following the procedure of Ma et al. (2025). LLMGrader evaluates an overall score along with five sub-scores: accuracy to prompt, creativity and originality, visual quality and realism, consistency and cohesion, and emotional or thematic resonance. We use the overall score as our optimization target.

**Datasets**   We consider two collections of simplified user prompts from the Lexica (Lexica, 2023) website and DiffusionDB (Wang et al., 2023b) for Promptist Reward, ImageReward and LLMGrader. We use the evaluation split used in Yun et al. (2025). For the DSG score, we use a subset of PartiPrompt (Yu et al., 2022) to follow the experimental setup of Mañas et al. (2024). We use a one-third subset of the evaluation prompts from each category (2D, 3D, and numeracy) in T2I-CompBench++ (Huang et al., 2025) for the UniDet evaluator[1].

---

[1]UniDet depends on T2I-CompBench++ dataset as it extracts desired image compositions from the prompt.

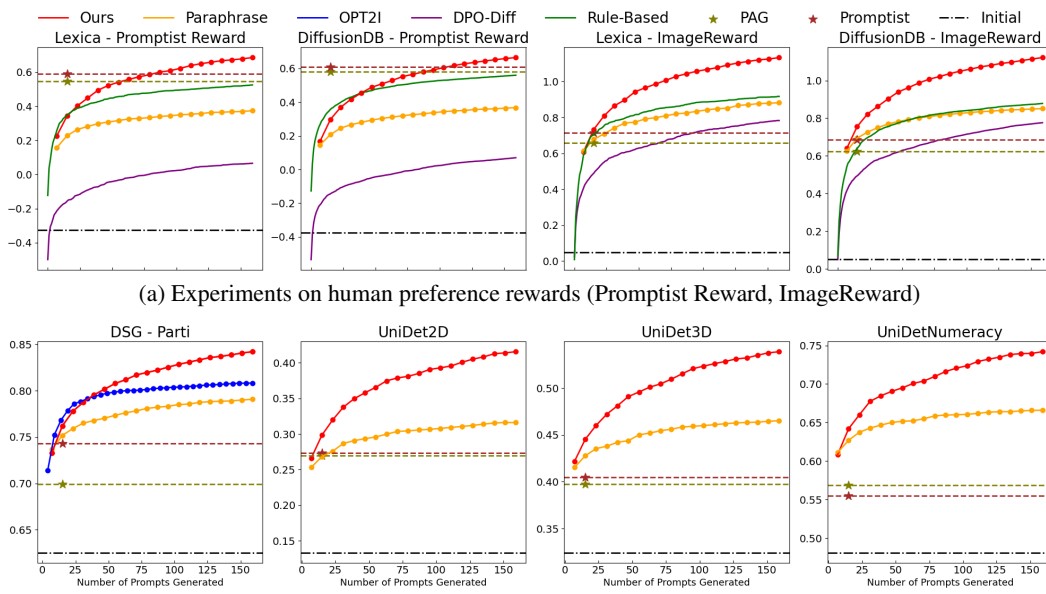

(a) Experiments on human preference rewards (Promptist Reward, ImageReward)

(b) Experiments on text-to-image consistency rewards (DSG, UniDet2D / 3D / Numeracy)

Figure 1: Optimization curves for human preference and text-to-image consistency rewards. Each curve shows how the reward changes as the number of generated prompts increases.

Table 1: Experimental results on LLMGrader reward in Lexica and DiffusionDB datasets. Test-time search methods are evaluated at the budget of 160 generated prompts. Full results in Tab. 14.

| Dataset | Method | Accuracy | Originality | Visual | Consistency | Emotional | Overall |
|---|---|---|---|---|---|---|---|
| Lexica | Initial | 67.85 | 62.84 | 80.11 | 83.93 | 69.59 | 72.49 |
| | Paraphrase | 69.73 | 67.24 | 83.94 | 86.40 | 73.92 | 89.01 |
| | **Ours** | **75.11** | **72.44** | **85.96** | **88.44** | **77.97** | **89.69** |
| DiffusionDB | Initial | 68.99 | 63.18 | 81.24 | 84.58 | 69.48 | 73.12 |
| | Paraphrase | 69.65 | 66.56 | 84.35 | 86.81 | 72.75 | 88.25 |
| | **Ours** | **73.39** | **70.66** | **86.40** | **88.30** | **76.14** | **88.99** |

**Baselines**  We compare RATTPO with both learning-based and test-time search-based methods. For the learning-based methods, we consider Promptist (Hao et al., 2023) and PAG (Yun et al., 2025) which train language models to directly enhance user-provided prompts. Following Yun et al. (2025), we generate sixteen responses from their official checkpoint by beam search and report the best one. Evaluations on reward setups other than the Promptist Reward are performed by applying them without retraining (*i.e.*, in a zero-shot manner).

For test-time search-based methods, we consider DPO-Diff (Wang et al., 2024), OPT2I (Mañas et al., 2024), and additionally two naive best-of-N baselines that modify the initial prompt by using LLM to paraphrase (denoted as Paraphrase) or in a rule-based manner (Rule-Based). Compared to RATTPO, the Paraphrase baseline does not use history or hint. The Rule-Based baseline randomly appends modifier words that are known to improve image aesthetics, similar to the heuristic used in Hao et al. (2023). Note that OPT2I is a reward-specific prompt engineer designed for text-to-image consistency scores, and thus the comparison is done only for the supported setup (DSG on PartiPrompt).

**Implementation Details**  We report averaged results across three runs with different seeds. Due to space constraints, we present the full results with standard deviations in App. F. We use the instruction-tuned `Gemma 3 27B` (Team et al., 2025) for all LLM components. We follow either PAG (Yun et al., 2025) or OPT2I (Mañas et al., 2024) for diffusion sampling hyperparameters. We conduct an ablation study on the choice of different-sized LLMs and different hyperparameters, such as diffusion sampler or history construction strategy for the hint-generator LLM, in App. D. To reduce the computational burden, we select two representative setups in our additional experiments: Promptist Reward and UniDet2D. We provide more implementation details in App. A.

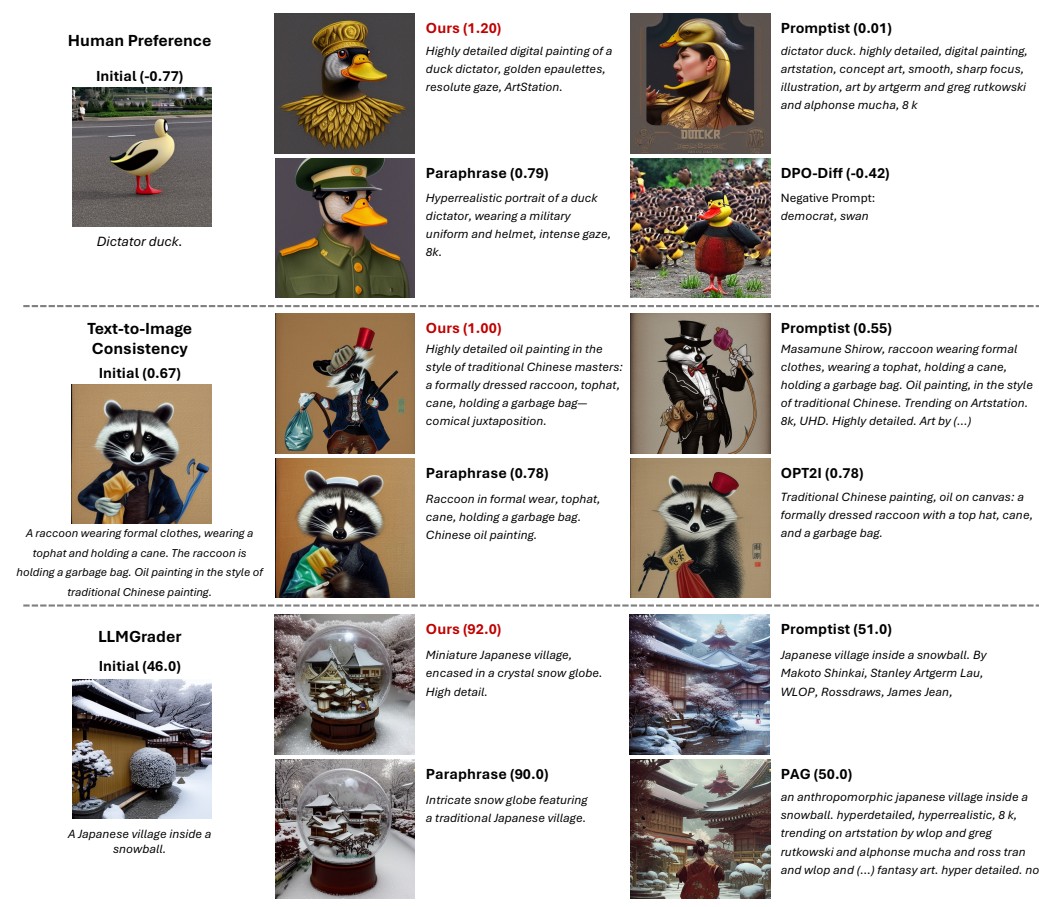

Figure 2: Qualitative results on diverse setups. As a reward-agnostic prompt engineer, RATTPO can enhance user prompts with respect to a broad range of reward functions.

## 4.2 MAIN RESULT

**Effectiveness of RATTPO for Diverse Rewards** As shown in Fig. 1 and Tab. 1, RATTPO consistently improves the initial prompts, with performance gains scaling with the inference budget. At an inference budget of 160 prompts, RATTPO achieves a significant improvement over the initial scores across all rewards and datasets. The qualitative results in Fig. 2 also showcase that RATTPO-optimized prompts can generate both aesthetically pleasing and correctly composed images depending on the target reward. Beyond human preference and image composition, RATTPO is also capable of improving user prompts to generate desired images on complex rubrics like originality and emotional resonance, when optimized against the overall LLMGrader score (Tab. 1).

As a test-time gradient-free prompt engineer, RATTPO effectively handles both differentiable and non-differentiable rewards. While previous learning-based methods and some test-time methods (*e.g.,* DPO-Diff) often rely on gradient-based optimization, many real-world rewards, such as the LLMGrader, are non-differentiable. Taken together, these results indicate that RATTPO is a reward-agnostic yet effective automated prompt engineer, capable of enhancing user prompts for a wide range of rewards.

**Comparison to Learning-Based Baselines** With sufficient inference budgets (~100 prompts), Fig. 1 shows that RATTPO can match the performance of learning-based methods on Promptist Reward without requiring expensive reward-specific training (which requires ~4 GPU days in Yun et al. (2025)). The result suggests a well-designed reward-agnostic prompt engineer can be as effective as its reward-specific counterparts, without the substantial cost of reward-specific training.

When we directly apply learning-based methods to unseen rewards, we observe significant performance drops. For instance, the results for ImageReward (Fig. 1a) show that their performance is only comparable to the simple Paraphrase baseline at the same inference cost, despite the correlation

Table 2: Results for the ablation study on hint. *Add. Hist.* denotes the variant using extra history instead of hint, and PR denotes Promptist Reward. RATTPO outperforms w/o Hint variants.

| Method | PR | UniDet2D |
|---|---|---|
| RATTPO | **0.683** | **0.416** |
| w/o Hint | 0.565 | 0.395 |
| w/o Hint + *Add. Hist.* | 0.522 | 0.387 |

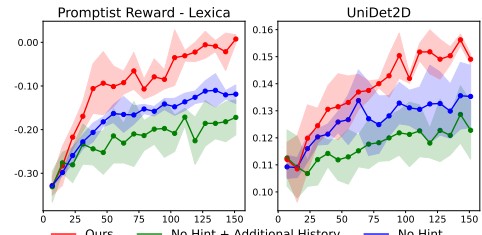

Figure 3: Per-iteration average reward plots for the hint ablation study.

between the trained and target rewards. The performance degradation is more severe in text-to-image consistency setups, where the correlation is weaker. These results indicate that the performance of learning-based methods drops quickly as the test reward diverges from the trained reward. This highlights the need for reward-agnostic prompt engineers: learning-based reward-specific prompt engineers require re-training for optimal performance, while reward-agnostic approaches like RATTPO can handle diverse reward functions that are defined and deployed at test-time.

**Comparison to Test-Time Search Baselines**   As shown in the optimization curve in Fig. 1, RATTPO outperforms other test-time search methods (DPO-Diff, OPT2I, Paraphrase, and Rule-Based) in terms of search efficiency. With an inference budget of 160 generated prompts, RATTPO surpasses all baselines across all experiment setups. When compared to the Rule-Based method, RATTPO shows a substantial performance advantage on ImageReward, while the advantage is smaller for Promptist Reward. This difference arises from the Rule-Based method's heuristics being specifically tailored for the Aesthetic Score component used in Promptist Reward. Consequently, the Rule-Based method performs only as well as simple paraphrasing on ImageReward, which highlights its limited generalizability across different reward. In contrast, RATTPO outperforms the Rule-Based method without relying on reward-specific heuristics, by progressively evolving prompt candidates using in-context learning from its optimization history.

When compared to OPT2I that employs an LLM as an optimizer with reward-specific meta-prompt (Fig. 1b), we observe that RATTPO achieves competitive scores at the early optimization stages but shows rapid improvement over time, leading to much higher reward at the end. We conjecture that the early saturation of OPT2I is due to heuristics in its design, which may contain an implicit inductive bias towards local optima. In contrast, RATTPO saturates slowly and achieves higher rewards, possibly by replacing reward-specific task description with hints, which are generated automatically from the optimization history and thus evolve as the optimization proceeds.

To further analyze the search efficiency in practice, we measure wall-clock time speedup of RATTPO over Paraphrase, the reward-agnostic baseline. Despite the overhead from an additional hint-generator LLM, RATTPO improves upon the Paraphrase baseline in terms of search efficiency, achieving up to $6.46\times$ wall-clock speedup as detailed in Tab. 3. On average, RATTPO is $4.81\times$ faster in end-to-end wall-clock time across our eight experimental setups, although specific speedups vary. Details about the search efficiency measurements are in App. C, along with the full results in Tab. 8.

Table 3: Search efficiency of RATTPO considering wall-clock time.

| | PR | UniDet2D |
|---|---|---|
| Search cost at win | 24 | 24 |
| Time, Paraphrase | 447s | 300s |
| Time, RATTPO at win | 69s | 51s |
| Speedup | **6.46×** | **5.90×** |

### 4.3 ANALYSIS

**Effectiveness of Hint**   To validate our hint mechanism, we compare RATTPO against two variants: one without hint and another replacing the hint generator with additional random history context. As shown in Tab. 2, removing hints degrades performance on both rewards, and simply adding more history alone does not compensate for this performance loss. Also, Fig. 3 depicts that variants without the hint achieve lower per-iteration average score. Based on the results, we claim that the hint mechanism in RATTPO effectively guides the optimizer LLM to produce better prompts every iteration, thereby improving the final reward. The performance gain from hint is not simply a

Table 4: Cross-reward experiment results. Underlined numbers indicate that RATTPO was not optimized for the evaluated reward.

| Optimization Target | Promptist Reward | UniDet2D |
|---|---|---|
| Initial (No Opt.) | -0.311 | 0.159 |
| RATTPO, optimize PR | 1.021 | 0.164 |
| RATTPO, optimize UniDet2D | -0.017 | 0.461 |

Table 5: Inter-reward hint transfer results.

| Method | PR | IR |
|---|---|---|
| RATTPO | **0.683** | **1.132** |
| w/o Hint | 0.565 | 1.081 |
| w/ IR Hint | 0.579 | - |
| w/ PR Hint | - | 1.065 |

**Initial Prompt**
playing guitar (-0.648)

**Example Context for Hint-Generator LLM**
Soulful, full-body portrait of a guitarist immersed in performance, highly detailed hands expertly playing, (...) emotive expression. (0.172)
Young adult playing electric guitar, stage lights, dynamic pose. (-0.660)

**Generated Hints**
Focus on full-body shots, (...) and consistently include "intricate hand positions" or "detailed hands on fretboard" alongside (...)

**Example Output from Optimizer LLM**
Passionate guitarist, full-body, intricate hand positions, Artgerm, WLOP, (...). (0.335)
A 26-year-old guitarist, full-body, intensely passionate performance, detailed hands, Artgerm, WLOP, dramatic lighting, 8k. (-0.148)

**Initial Prompt**
a couch on the left of a dog (0.197)

**Example Context for Hint-Generator LLM**
A detailed living room scene: a couch on the left, a dog on the right. (0.281)
A comfortable couch is to the left of a happy dog. (0.0)

**Generated Hints**
Focus on detailed descriptions of the living room, couch, and dog (breed, color, texture) while maintaining clear left/right positioning and emphasizing realism/quality.

**Example Output from Optimizer LLM**
Realistic living room: a linen sectional couch (left) and a relaxed, cream-colored Labrador retriever (right), soft lighting. (0.303)
Cozy living room, left: a velvet teal couch, right: a golden retriever. (0.229)

(a) Human Preference (Promptist Reward)     (b) Text-to-Image Consistency (UniDet2D)

Figure 4: Case study of generated hints. Numbers in parentheses indicate reward for corresponding prompts. The Hint-generator LLM summarizes the search history to generate a "hint" that instructs the optimizer LLM. We highlight the relevant parts and omit (...) some words for better presentation.

consequence of having more history in the context window, but rather the result of an well-designed mechanism for aggregating history to explicitly guide the search.

To further evaluate whether hint contains reward-specific optimization signal, we design an inter-reward hint transfer experiment. Specifically, we consider variants that use pre-generated hints from other reward setup instead of generating hints during optimization. For instance, w/IR Hint variant optimizes for Promptist Reward using hints generated by RATTPO during ImageReward optimization. Since hint is generated per-prompt, we consider the transfer within the same dataset (Lexica). As shown in Tab. 5, these variants show degraded performance, and do not meaningfully improve upon RATTPO without the hint. Thus, we claim that hints indeed contain reward-specific information, thanks to our explicit design that queries the hint-generator LLM about optimization strategy.

Lastly, we conduct a case study on generated hints, exploiting their human interpretability (Fig. 4, see App. H for more examples). As can be seen, the hint-generator LLM is capable of recognizing high-scoring patterns and summarizing them as textual descriptions. For instance, highlighted parts in Fig. 4a shows that the hint-generator LLM instructs the optimizer LLM to focus on "full-body portrait" and "detailed hand positions", which can be inferred by comparing high-scoring prompts with low-scoring prompts. Also, by comparing the generated hints for different rewards, we observe that the hint-generator can capture the information about the underlying reward function. The hint-generator instructs the optimizer LLM to add fine-grained details for improving image aesthetics (Fig. 4a), and to explicitly mention both 'left' and 'right' keywords for better drawing images with spatial object relationships (Fig. 4b). Hints are not only an effective way to capture reward-aware optimization signals, but they also provide a means to analyze the optimization process.

**Cross-Reward Evaluation** While we extensively verify the effectiveness of RATTPO in optimizing target rewards, it would be undesirable if the improved reward comes at the cost of overall image quality. To validate that RATTPO does not fall into such an over-optimization scenario, we conduct a cross-reward experiment. Specifically, we optimize prompts for human preference (Promptist

Table 6: Experimental results with various diffusion backbones. Asterisk denotes the baselines trained for Promptist Reward. Test-time search methods are evaluated with a 160-prompt budget.

| Method | Human Preference | | | | Text–to-Image Consistency | | | |
|---|---|---|---|---|---|---|---|---|
| | SD1.4 | SD2.1 | SDXL-Turbo | FLUX | SD1.4 | SD2.1 | SDXL-Turbo | FLUX |
| Initial | -0.325 | -0.373 | -0.327 | -0.480 | 0.123 | 0.133 | 0.162 | 0.240 |
| Promptist* | 0.591 | 0.345 | 0.257 | 0.104 | 0.255 | 0.273 | 0.322 | 0.413 |
| PAG* | 0.545 | 0.219 | 0.193 | -0.013 | 0.275 | 0.326 | 0.327 | 0.439 |
| DPO-Diff | 0.066 | -0.037 | - | - | - | - | - | - |
| Paraphrase | 0.372 | 0.261 | 0.415 | 0.138 | 0.384 | 0.316 | 0.419 | 0.536 |
| **Ours** | **0.683** | **0.503** | **0.487** | **0.445** | **0.396** | **0.416** | **0.454** | **0.578** |

Table 7: Results of RATTPO combined with test-time alignment (DAS (Kim et al., 2025)). PR denotes Promptist Reward.

| Method | PR |
|---|---|
| Initial | -0.325 |
| +DAS | 0.411 |
| +DAS + RATTPO | **1.006** |

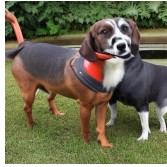 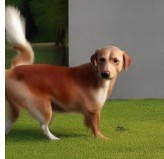 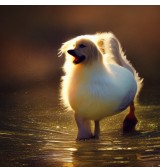

(a) Initial     (b) DAS     (c) DAS+RATTPO

Figure 5: Qualitative examples of combining RATTPO with DAS (Kim et al., 2025). Prompts are in App. E

Reward) and evaluate them on an uncorrelated reward for spatial composition (UniDet2D), and vice versa. In Tab. 4, we observe that the non-optimized rewards (underlined) remain at a similar level or even slightly improved compared to their initial scores. This result suggests that RATTPO does not over-optimize prompts for the target reward. We hypothesize that RATTPO is robust to such over-optimization for two main reasons. First, our optimization is performed in a discrete text space without gradients, which may act as an implicit form of regularization. Second, we explicitly instruct the LLM to restrict its search space to prompts that preserve the semantics of the initial prompt. This constraint makes over-optimization scenarios with degraded faithfulness less feasible.

**Robustness to Diffusion Backbone**  As a test-time search method, RATTPO is also agnostic to the choice of T2I models. To validate this claim, we experiment with various T2I diffusion backbones and report the results in Tab. 6. Specifically, we consider SDXL-Turbo (Sauer et al., 2024) and FLUX.1 Schnell (Black Forest Labs, 2024) in addition to Stable Diffusion 1.4 and 2.1 that are used in our main experiments. The results show that RATTPO is robust to the choice of diffusion backbone, consistently improving the initial prompt by a large margin. We also note that not all test-time methods are agnostic to diffusion backbone, exemplified by DPO-Diff (Wang et al., 2024). DPO-Diff optimizes a negative prompt for improving image, where negative prompts are often not used in timestep-distilled models (SDXL-Turbo and FLUX.1 Schnell). In contrast, RATTPO directly optimizes the user prompt and is applicable to these models.

**RATTPO with Test-Time Alignment**  Since RATTPO is agnostic to denoising process, it is compatible with test-time alignment methods that tweak sampling steps. For instance, RATTPO can be combined with DAS (Kim et al., 2025), a sampling-as-alignment approach that effectively aligns a diffusion model without training. We report the experimental results on Promptist Reward in the Lexica dataset in Tab. 7 and Fig. 5. The results demonstrate that our automated prompt engineer can be easily combined with other test-time alignment methods to generate images with higher rewards.

## 5 CONCLUSION

We introduce RATTPO, a reward-agnostic prompt optimization framework that can be applied to diverse reward functions without task-specific modifications. Our approach employs an iterative optimization process using two complementary LLMs: the optimizer LLM to propose enhanced prompts and the hint-generator LLM to provide contextual feedback based on reward signals. RATTPO is a reward-agnostic yet effective prompt engineer, thanks to replacing the reward-specific task description in the meta-prompt with a hint that is automatically generated by an LLM on-the-fly. We validate the effectiveness of RATTPO in various reward setups, categorized by human preference, text-to-image

consistency, and holistic MLLM assessment. Experimental results demonstrate that RATTPO is applicable to a wide range of rewards and surpasses other test-time approaches in terms of search efficiency.

## ETHICS STATEMENT

We have carefully reviewed the Code of Ethics and confirm that we adhere to the principles. To the best of our knowledge, this work raises no ethical concerns. The use of LLMs in this paper is clarified in App. I.

## REPRODUCIBILITY STATEMENT

We have made our best efforts to ensure the reproducibility of our experiments. We include the code in our submission to enable others to replicate our results. We report all implementation details (App. A) for the experiments, including meta-prompts used for LLMs (App. B). We expect the numbers in main results to be reproducible, as we report the average of three different runs. Except the LLMGrader experiment, we use publicly available datasets and open-sourced models. For dataset, exact lists/splits and preprocessing scripts for the datasets are included in our attached code.

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

APPENDIX

# A    IMPLEMENTATION DETAILS

**Sampling from Diffusion Backbone**    Following PAG (Yun et al., 2025), we use Stable Diffusion 1.4 (Rombach et al., 2022) with DPM solver (Lu et al., 2022) as a default diffusion sampling setup. The inference is done in half precision (`fp16`). For text-to-image consistency reward experiments, we follow OPT2I (Mañas et al., 2024) and use Stable Diffusion 2.1 (Rombach et al., 2022) as a diffusion backbone. We set the number of inference steps to 20 by default. Furthermore, for the diffusion backbone robustness experiment, SDXL-Turbo (Sauer et al., 2024) and FLUX.1 Schnell (Black Forest Labs, 2024) are timestep distilled models, and thus we use the PNDM (Liu et al., 2022) solver and Euler solver with the number of inference steps set to one and four, respectively. For the experiment combining RATTPO with test-time alignment method, DAS (Kim et al., 2025), we use DDIM (Song et al., 2021) solver, as the official implementation[1] is provided only for DDIM sampler. We also evaluate RATTPO with various diffusion samplers (Tab. 11) and find that it is robust to different sampler choices.

**Hyperparameters**    For RATTPO, we perform 20 iterations where each iteration proposes 8 prompts. We select the 8 best prompts from the history for the context of the optimizer LLM, and 20 random prompts for the hint-generator LLM. We use default hyperparameters when querying the LLM. For baselines, we use the default setup if possible. To match the inference cost (*i.e.*, 160 generated prompts), we increase the length of the evolutionary search phase in DPO-Diff, resulting in 20 iterations of gradient-based optimization and 140 iterations of evolutionary search. The evolutionary search phase may terminate early if the search space contains fewer than 140 candidates. For OPT2I (Mañas et al., 2024), we increase the iteration, resulting in 32 optimization iterations where each iteration generates 5 prompts. For the experiment combining RATTPO with DAS (Kim et al., 2025), the hyperparameters for DAS are set to default ($\gamma = 0.008$, $\alpha = 0.005$).

**Rewards**    In our experiment, we consider various rewards that can be categorized into (1) Human Preference Reward (2) Text-Image Consistency Reward (3) Holistic MLLM Assessment. Except for the LLMGrader used for the last category, all other models are publicly available. We employ widely adopted reward models for each category, namely Promptist Reward (Hao et al., 2023), ImageReward (Xu et al., 2023), DSG Score (Cho et al., 2024), and UniDet (Zhou et al., 2022) scorer from T2I-Compbench (Huang et al., 2025) benchmark. For LLMGrader, we follow the setup in Ma et al. (2025); the prompt used for querying LLM is shown in Fig. 6. When calculating rewards, we follow Yun et al. (2025) to estimate the expectation in Eq. 1 by averaging rewards over generated images from three different initial diffusion noise samples. The initial noises are shared across methods and iterations.

**Datasets**    For the human preference and LLMGrader experiments, we directly use the evaluation splits of Lexica (Lexica, 2023) and DiffusionDB (Wang et al., 2023b) used in PAG (Yun et al., 2025). The datasets consist of 64 and 256 preprocessed prompts, respectively, obtained by extracting the main content from human-engineered prompts. For PartiPrompt (Yu et al., 2022) dataset used for DSG (Cho et al., 2024) reward setup, we prepare the dataset following (Mañas et al., 2024). Specifically, we use the first 50 prompts from four categories ("Properties and Positioning", "Quantity", "Fine-grained Detail", "Complex"). As the category "Properties and Positioning" contains only 35 prompts, the resulting dataset has 185 initial prompts in total. Lastly, for UniDet-based evaluators, we use accompanying datasets used in T2I-Compbench++ (Huang et al., 2025) benchmark. Each validation set for 2D, 3D and Numeracy category includes 300 prompts, and we use subset of 100 prompts in each category.

**Baselines**    We provide some additional details for baselines used in our experiments.

- DPO-Diff (Wang et al., 2024) baseline in our experiment uses the default setting recommended in the original paper. Specifically, we optimize negative prompts with hybrid (gradient-based

---

[1]https://github.com/krafton-ai/DAS

You are a multimodal large-language model tasked with evaluating images generated by a text-to-image model.
Your goal is to assess each generated image based on specific aspects and provide a detailed critique, along with a scoring system.
The final output should be formatted as a JSON object containing individual scores for each aspect and an overall score.
Below is a comprehensive guide to follow in your evaluation process:
1. Key Evaluation Aspects and Scoring Criteria:
For each aspect, provide a score from 0 to 100, where 0 represents poor performance and 100 represents excellent performance.
For each score, include a short explanation or justification (1-2 sentences) explaining why that score was given.
The aspects to evaluate are as follows:
a) Accuracy to Prompt
Assess how well the image matches the description given in the prompt.
Consider whether all requested elements are present and if the scene, objects, and setting align accurately with the text.
Score: 0 (no alignment) to 100 (perfect match to prompt).
Creativity and Originality
Evaluate the uniqueness and creativity of the generated image.
Does the model present an imaginative or aesthetically engaging interpretation of the prompt?
Is there any evidence of creativity beyond a literal interpretation?
Score: 0 (lacks creativity) to 100 (highly creative and original).
c) Visual Quality and Realism
Assess the overall visual quality, including resolution, detail, and realism.
Look for coherence in lighting, shading, and perspective.
Even if the image is stylized or abstract, judge whether the visual elements are well-rendered and visually appealing.
Score: 0 (poor quality) to 100 (high-quality and realistic).
d) Consistency and Cohesion
Check for internal consistency within the image.
Are all elements cohesive and aligned with the prompt?
For instance, does the perspective make sense, and do objects fit naturally within the scene without visual anomalies?
Score: 0 (inconsistent) to 100 (fully cohesive and consistent).
e) Emotional or Thematic Resonance
Evaluate how well the image evokes the intended emotional or thematic tone of the prompt.
For example, if the prompt is meant to be serene, does the image convey calmness?
If it's adventurous, does it evoke excitement?
Score: 0 (no resonance) to 100 (strong resonance with the prompt's theme).
2. Overall Score
After scoring each aspect individually, provide an overall score, representing the model's general performance on this image.
This should be a weighted average based on the importance of each aspect to the prompt or an average of all aspects.Now grade the image based on the above criteria.
Below is the prompt:
Prompt: {prompt}

Figure 6: The prompt used for LLMGrader reward. We follow the prompt used in Ma et al. (2025).

optimization with evolutionary search) search. Note that this setup is not applicable to both (1) non-differentiable rewards in text-to-image consistency and MLLM score, and (2) diffusion backbones like SDXL-Turbo (Sauer et al., 2024) or FLUX.1 Schnell (Black Forest Labs, 2024) that do not use negative prompt. We exclude DPO-Diff baseline for such setups.

- OPT2I (Mañas et al., 2024) baseline in our experiment is implemented by using the exact hyperparameters and meta-prompt following the original paper. We use the same LLM (Gemma 3 27B) as RATTPO for fair comparison. While the paper claims that OPT2I can be used for arbitrary text-to-image consistency score, it uses different meta-prompts for each choice of the score. Among the experimental setups used in our paper, the DSG reward on PartiPrompt is the only setup for which the meta-prompt for OPT2I is provided, and thus we include OPT2I as a baseline only in that setup. As Gemma 3 does not use a system role, we include the system prompt in the user prompt.

> As an expert prompt engineer for text-to-image generation, rewrite the original prompt in 8 distinct ways to improve the visual quality of the resulting images.
> To aid you in this task, you will be also given 8 history prompts that are already tried before. For each history prompt, its score is given as number. Higher score indicates that the prompt is better.
> (Hint: {hint})
> You can use the scores to guide your rewriting process and thus improve the visual quality of the generated image, but your response should be different from histories.
> Histories:
> 1. Prompt: {prompt_1} (Score: {score_1})
> 2. Prompt: {prompt_2} (Score: {score_2})
> (...)
> 8. Prompt: {prompt_8} (Score: {score_8})
> Return exactly 8 variations, numbered 1 through 8, each on its own line and ordered from shortest to longest.
> Preserve the meaning of the original prompt and keep each variation under 70 words. Start your output immediately with the numbered variations.
> Original Prompt: {initial_prompt}

Figure 7: Meta-prompt used for the optimizer LLM. We highlight the history-related part and hint-related part in colored texts.

- The Rule-Based baseline is constructed in a similar way to the heuristic baseline used in Promptist (Hao et al., 2023). Specifically, we collect the top 15 most frequent words appearing in human-engineered prompts, and randomly append three of them to the initial prompt. The word pool consists of: `"concept art"`, `"highly detailed"`, `"sharp focus"`, `"artstation"`, `"digital painting"`, `"intricate"`, `"illustration"`, `"trending on artstation"`, `"smooth"`, `"elegant"`, `"octane render"`, `"fantasy"`, `"wlop"`, `"digital art"`, and `"8 k"`. As this heuristic rule is targeted for enhancing image aesthetics, we only include it for experiments that contain human preference.

**Computational Resources** We conduct the experiments with our internal GPU servers that consist of two types of machines. We list their specifications below.

1. Intel Xeon Gold 6330 CPU and NVIDIA RTX A6000 GPU (with 48GB VRAM)

2. Intel Xeon Gold 6230 CPU and NVIDIA RTX 3090 GPU (with 24GB VRAM)

For LLM components, we either use (1) API provided by Google Generative AI service[2] (`gemma-3-27b-it`) or (2) self-hosted LLM server constructed with ollama[3] (`gemma3:27b`) for using `Gemma`. When implementing LLMGrader with `Gemini-1.5-Flash` (Team et al., 2024), we use the official API[4].

Total running time depends on the GPU machine and also LLM API response time. When using the machine with A6000 GPU and using Google GenAI API for LLM inference, we expect a single optimization (20 iterations for one initial prompt) to take about 10 minutes.

## B META-PROMPT

Meta-prompts used for querying the optimizer LLM and the hint-generator LLM can be found in Fig. 7 and Fig. 8, respectively. Note that history and hint are not available for the first search iteration in RATTPO. In this case, the meta-prompt for the optimizer LLM reduces to the prompt for Paraphrase baseline, which uses neither optimization history nor hint. This corresponds to a prompt that consists of uncolored texts in Fig. 7.

---

[2] https://ai.google.dev/gemma/docs/core/gemma_on_gemini_api

[3] https://ollama.com/library/gemma3

[4] https://ai.google.dev/gemini-api/docs/models#gemini-1.5-flash

> As an expert prompt engineer for text-to-image generation, you are trying to rewrite the original prompt to improve the scores. Below are some histories of prompts you tried before. Based on the history prompts with corresponding scores, guess how you can enhance the score.
> Histories:
> Prompt: {`prompt_1`} (Score: {`score_1`})
> Prompt: {`prompt_2`} (Score: {`score_2`})
> (...)
> Prompt: {`prompt_20`} (Score: {`score_20`})
>
> Now describe the way how we can increase the score in plain words. Simply output the way in a single line.

Figure 8: Meta-prompt used for the hint-generator LLM.

Table 8: Wall-clock time analysis of search efficiency, compared to Paraphrase baseline

| | Lexica | | DiffusionDB | | Parti | UniDet (reward) | | |
| --- | --- | --- | --- | --- | --- | --- | --- | --- |
| | PR | IR | PR | IR | DSG | 2D | 3D | Numeracy |
| Average # of prompts at win | 24 | 40 | 24 | 32 | 40 | 24 | 40 | 32 |
| Wall clock time, Paraphrase (s) | 447 | 383 | 410 | 355 | 1151 | 300 | 328 | 310 |
| Wall clock time, RATTPO at win (s) | 69 | 106 | 62 | 82 | 300 | 51 | 94 | 74 |
| Speedup ($\times$) | 6.46 | 3.62 | 6.66 | 4.34 | 3.84 | 5.90 | 3.48 | 4.20 |

Table 9: Ablation study on LLM capacity. Bold denotes our default setting.

| Method | Promptist Reward | UniDet2D |
| --- | --- | --- |
| Initial | -0.325 $\pm$0.013 | 0.133 $\pm$0.007 |
| Gemma 3 1B | 0.558 $\pm$0.019 | 0.401 $\pm$0.002 |
| Gemma 3 12B | 0.563 $\pm$0.009 | 0.415 $\pm$0.011 |
| **Gemma 3 27B** | **0.683 $\pm$0.090** | **0.416 $\pm$0.015** |

## C   SEARCH EFFICIENCY OF RATTPO

For a practical analysis of search efficiency, we measure the wall-clock time it takes RATTPO to achieve the peak score of the Paraphrase baseline. Specifically, we first record the average reward and wall-clock time of the Paraphrase baseline at a fixed prompt budget of 160. For RATTPO, we then track the average reward and cumulative runtime at each search round, again averaged over all runs. We identify the earliest search round at which RATTPO's average reward meets or exceeds the Paraphrase baseline's average reward, and the corresponding prompt budget is reported (Average # of prompts at win). The associated runtime (Time, RATTPO at win) is used to calculate the speedup in the table, which is the ratio between the baseline runtime and this RATTPO runtime.

We extend the analysis in Tab. 3 by averaging the results across all eight experimental setups. As detailed in Tab. 8, RATTPO achieves a net wall-clock speedup of $4.81\times$ over the Paraphrase baseline on average.

## D   ADDITIONAL EXPERIMENT RESULTS

**Ablation on LLM Size**    RATTPO relies on the ICL ability of LLMs for prompt optimization, where larger LLMs are often better at (Brown et al., 2020). To see how the optimization performance changes with the choice of different-sized LLM, we experiment with 1B, 12B and 27B variants of `Gemma 3` (Team et al., 2025). Results in Tab. 9 shows that RATTPO, even with the smallest size 1B variant, can yield meaningful improvements over the initial prompt. We use 27B model as default setting, which works the best in both settings.

**Ablation on Hint-Generator LLM Context**    For the hint-generator LLM, we sample 20 random history prompts and scores to use for its context. Tab. 10 shows an ablation study for this choice. For

Table 10: Ablation study on hint history selection. Bold denotes our default setting.

| Selection Strategy | Promptist Reward | UniDet2D |
|---|---|---|
| **Random** | **0.683** ±**0.090** | **0.416** ±**0.015** |
| Best | 0.614 ±0.045 | 0.411 ±0.013 |
| Number of History | Promptist Reward | UniDet2D |
| 0 (w/o hint) | 0.565 ±0.033 | 0.395 ±0.038 |
| 4 | 0.674 ±0.030 | 0.409 ±0.006 |
| **20** | **0.683** ±**0.090** | **0.416** ±**0.015** |
| All | 0.557 ±0.023 | 0.406 ±0.002 |

Table 11: Experiment results with various diffusion samplers. Asterisk denotes learning baselines trained for Promptist Reward. Test-time search methods are evaluated with a 160-prompt budget.

| | PR+Lexica | | | UniDet2D | | |
|---|---|---|---|---|---|---|
| **Sampler** | **DPM** | **DDIM** | **PNDM** | **DPM** | **DDIM** | **PNDM** |
| Initial | -0.308 | -0.339 | -0.404 | 0.125 | 0.116 | 0.101 |
| Promptist* | 0.540 | 0.571 | 0.518 | 0.279 | 0.272 | 0.272 |
| PAG* | 0.582 | 0.525 | 0.436 | 0.267 | 0.261 | 0.238 |
| Paraphrase | 0.374 | 0.509 | 0.473 | 0.325 | 0.389 | 0.369 |
| **Ours** | **0.662** | **0.644** | **0.594** | **0.429** | **0.430** | **0.382** |

the history selection strategy, we evaluate three methods: (1) selecting random histories, (2) using a mix of the best and worst histories, and (3) selecting only the best histories. For each method, we also experiment with varying the number of histories provided as context. As can be seen, our default setting performs the best in both setups.

**Ablation on Diffusion Sampler**   As a test-time optimization method, RATTPO is robust to different diffusion samplers. To empirically support this claim, we additionally evaluate the performance of RATTPO using two different samplers implemented in the diffusers library (DDIM and PNDM). As shown in Tab. 11, our method is robust to the choice of diffusion sampler, effectively improving user prompts across various sampler configurations. As a test-time reward-agnostic prompt engineer, RATTPO is compatible with both a broad set of rewards and diverse generation setups, including different diffusion models (Tab. 6) and samplers (Tab. 11).

## E   FULL PROMPTS IN DAS EXPERIMENT

For the qualitative sample in Fig. 5, we use a prompt from Lexica dataset. The initial prompt is "Duck and dog crossbreed", and the RATTPO-optimized prompt is "Captivating digital art: a duck-dog hybrid with flowing fur transitioning to feathers, bathed in golden light, emphasizing an intelligent, inquisitive gaze".

## F   FULL MAIN RESULTS TABLE WITH STANDARD DEVIATION

As we omit the standard deviations in main text (due to spatial constraint), we present the full results here. See Tab. 12-15.

## G   ADDITIONAL QUALITATIVE RESULTS

Additional qualitative examples, for all combinations of rewards and datasets in our main experiment, can be found in Fig. 9-13.

Table 12: Full results for human preference reward experiments. Asterisk denotes learning baselines trained for Promptist Reward. Test-time search methods are evaluated at the budget of 160 generated prompts.

| Method | Promptist Reward | | ImageReward | |
|---|---|---|---|---|
| | Lexica | DiffusionDB | Lexica | DiffusionDB |
| Initial | -0.325 ±0.013 | -0.375 ±0.001 | 0.049 ±0.143 | 0.052 ±0.096 |
| Promptist* | 0.591 ±0.034 | 0.609 ±0.020 | 0.714 ±0.114 | 0.686 ±0.108 |
| PAG* | 0.545 ±0.014 | 0.581 ±0.002 | 0.657 ±0.144 | 0.623±0.103 |
| DPO-Diff | 0.066 ±0.009 | 0.070 ±0.006 | 0.783 ±0.024 | 0.775 ±0.060 |
| Paraphrase | 0.372 ±0.004 | 0.365 ±0.013 | 0.880 ±0.170 | 0.850 ±0.062 |
| **Ours** | **0.683±0.090** | **0.663±0.020** | **1.132±0.049** | **1.121±0.036** |

Table 13: Full results for text-to-image consistency reward experiments. Asterisk denotes learning baselines trained for Promptist Reward. Test-time search methods are evaluated at the budget of 160 generated prompts.

| Method | DSG | UniDet | | |
|---|---|---|---|---|
| | Parti | 2D | 3D | Numeracy |
| Initial | 0.625 ±0.013 | 0.133 ±0.007 | 0.324 ±0.005 | 0.481 ±0.004 |
| Promptist* | 0.743 ±0.010 | 0.273 ±0.008 | 0.405±0.013 | 0.555±0.007 |
| PAG* | 0.693 ±0.005 | 0.269 ±0.002 | 0.397±0.007 | 0.569±0.004 |
| Paraphrase | 0.791 ±0.007 | 0.316 ±0.008 | 0.465 ±0.009 | 0.666±0.004 |
| **Ours** | **0.842 ±0.005** | **0.416 ±0.015** | **0.539 ±0.009** | **0.741 ±0.006** |

Table 14: Full results for LLMGrader reward experiments in Lexica dataset. Asterisk denotes learning baselines trained for Promptist Reward. Test-time search methods are evaluated at the budget of 160 generated prompts.

| Method | Accuracy | Originality | Visual | Consistency | Emotional | Overall |
|---|---|---|---|---|---|---|
| Initial | 67.8 ±1.12 | 62.8±1.52 | 80.1±0.41 | 83.9±0.66 | 69.6±1.39 | 72.5±0.85 |
| Promptist* | 60.5 ±1.94 | 64.7 ±1.29 | 85.3 ±0.52 | 85.9 ±0.18 | 68.9 ±0.85 | 85.8 ±1.57 |
| PAG* | 57.7 ±2.03 | 64.2 ±2.46 | 84.7 ±0.96 | 84.8 ±0.39 | 68.2 ±1.56 | 85.2 ±1.61 |
| Rule-Based | 68.4 ±2.21 | 66.0 ±2.68 | 83.8 ±1.97 | 85.4 ±1.56 | 71.5 ±2.61 | 88.4 ±0.87 |
| Paraphrase | 69.7 ±2.68 | 67.2 ±1.24 | 83.9 ±0.70 | 86.4 ±1.23 | 73.9 ±2.22 | 89.0 ±0.44 |
| **Ours** | **75.1±2.49** | **72.4 ±1.52** | **86.0 ±0.65** | **88.4 ±0.09** | **78.0±1.30** | **89.7±0.40** |

Table 15: Full results for LLMGrader reward experiments in DiffusionDB dataset. Asterisk denotes learning baselines trained for Promptist Reward. Test-time search methods are evaluated at the budget of 160 generated prompts.

| Method | Accuracy | Originality | Visual | Consistency | Emotional | Overall |
|---|---|---|---|---|---|---|
| Initial | 69.0±0.13 | 63.2±0.16 | 81.2±0.12 | 84.6±0.02 | 69.5±0.22 | 73.1±0.13 |
| Promptist* | 59.1±1.69 | 65.9±1.37 | 86.4±0.84 | 86.3±0.32 | 68.8±1.04 | 84.6±1.13 |
| PAG* | 52.5±0.38 | 62.9±0.22 | 85.7±0.51 | 84.7±0.13 | 64.2±0.22 | 83.6±0.04 |
| Rule-Based | 68.1±1.95 | 67.0±2.41 | 84.9±0.96 | 86.6±0.83 | 71.3±2.13 | 87.2±0.58 |
| Paraphrase | 69.7 ±1.00 | 66.6 ±0.77 | 84.3 ±0.46 | 86.8 ±0.53 | 72.8 ±0.86 | 88.3 ±0.31 |
| **Ours** | **73.4±1.44** | **70.7±1.33** | **86.4 ±0.51** | **88.3±0.40** | **76.1 ±1.15** | **89.0±0.33** |

# H  ADDITIONAL HINT CASE STUDY

Additional hint case study, for all combinations of rewards and datasets in our main experiment, can be found in Fig. 14-23. Similar to the figure in the main text, We highlight the relevant parts.

Also note that the hint sometimes suggests avoiding phrases that have negative effect, visualized by underlined text. Numbers in parentheses indicate reward for corresponding prompts.

## I USE OF LLMS

We utilize LLMs to check grammar and improve the clarity of the sentences in this paper. The authors draft all the original content, and the role of the LLMs is strictly limited to language refinement. All final contents are carefully verified by the authors.

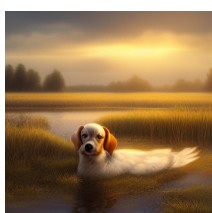

**Initial (−1.49)**

*Duck and dog crossbreed*

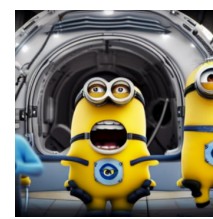

**Initial (−0.43)**

*a huddle of astronauts dissecting a minion from despicable me.*

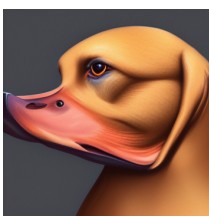

**Ours (1.44)**

*Immersive digital painting: A realistic duck-dog hybrid peacefully stands in a misty wetland, bathed in the warm glow of golden hour. Focus on detailed textures, atmospheric perspective, and soft, natural lighting.*

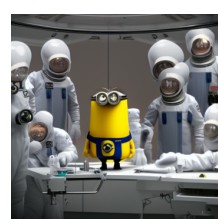

**Ours (0.60)**

*Ultra-detailed: astronauts performing a precise dissection of a Minion in a sterile laboratory, dramatic volumetric lighting, ArtStation.*

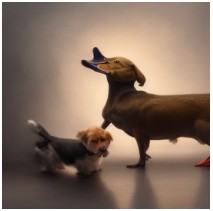

**Paraphrase (0.96)**

*Highly detailed digital painting of a duck-dog hybrid, combining canine and avian features, dramatic lighting, artstation.*

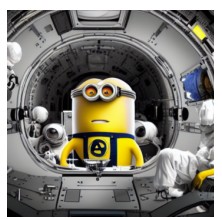

**Paraphrase (0.47)**

*Highly detailed image: astronauts dissecting a Minion, sterile lab environment.*

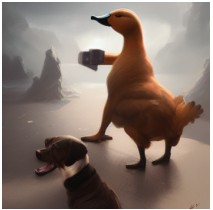

**PAG (0.41)**

*Duck and dog crossbreed, hyperrealistic, cinematic, dramatic lighting, by wlop*

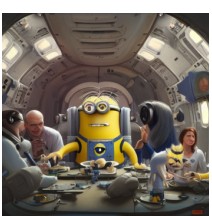

**PAG (-3.40)**

*hyperrealistic, hyperdetailed, 3 d render, 8 k resolution, by wlop and greg rutkowski*

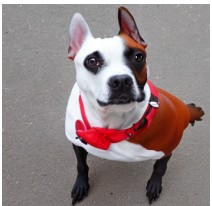

**Promptist (-0.54)**

*Duck and dog crossbreed, painting by Craig Mullins, 4k. trending on artstation.*

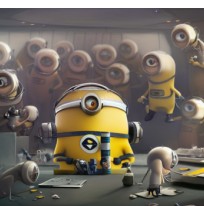

**Promptist (-0.06)**

*an a huddle of astronauts dissecting a minion from despicable me, by greg rutkowski, by wlop, by namek*

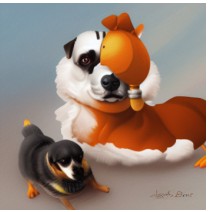

**DPO-Diff (-1.84)**

*Negative Prompt: lion,but,snake,separate*

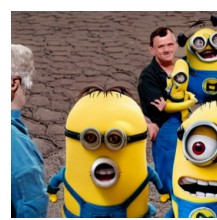

**DPO-Diff (-0.67)**

*Negative Prompt: separate,for,passengers,building,individual,to,her*

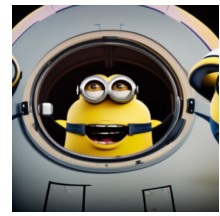

**Rule-Based (0.79)**

*Duck and dog crossbreed, illustration, fantasy, smooth*

**Rule-Based (-0.10)**

*a huddle of astronauts dissecting a minion from despicable me, 8 k, sharp focus, elegant.*

(a) Promptist Reward - Lexica                    (b) Promptist Reward - DiffusionDB

Figure 9: Additional qualitative results for Promptist Reward in Lexica (left) and DiffusionDB (right) datasets.

**Initial (−0.27)**

*an intricate concept illustration of bird landing*

**Initial (−0.25)**

*halloween witch girl with making witchcraft.*

**Ours (1.61)**

*Highly detailed concept art: a small bird, wings partially folded, landing gently on a vibrant meadow of wildflowers, rendered in a painterly style with dramatic golden hour lighting.*

**Ours (1.80)**

*Halloween witch girl meticulously crafting a magical potion, vibrant ingredients, atmospheric and warm lighting, detailed digital painting.*

**Paraphrase (0.69)**

*A breathtaking concept illustration of a bird landing – highly detailed, painterly style, dramatic lighting.*

**Paraphrase (1.10)**

*Atmospheric Halloween scene: a young witch girl intensely focused on making witchcraft, glowing potion, dark fantasy, intricate details, cinematic lighting.*

**PAG (1.31)**

*an intricate concept illustration of bird landing, by wlop, by wlop, by ilyakuvshinov, by greg rutkowski*

**PAG (0.75)**

*halloween witch girl with making witches. hyper-detailed, 8k, HD. by wlop.*

**Promptist (0.86)**

*an intricate concept illustration of bird landing, by greg rutkowski, artstation, cgsociety, dramatic lighting, highly detailed, incredible quality, trending on artstation*

**Promptist (0.31)**

*Halloween witch girl with making witches. , digital Art , WLOP, Rossdraws, James Jean,*

**DPO-Diff (0.32)**

*Negative Prompt: none,simple,fact,abstraction ,none,reptile*

**DPO-Diff (0.04)**

*Negative Prompt: saint,elder,without,destroyi ng,purity*

**Rule-Based (1.02)**

*an intricate concept illustration of bird landing, wlop, fantasy, digital art*

**Rule-Based (0.98)**

*halloween witch girl with making witchcraft, digital art, octane render, digital painting.*

(a) ImageReward - Lexica        (b) ImageReward - DiffusionDB

Figure 10: Additional qualitative results for ImageReward in Lexica (left) and DiffusionDB (right) datasets.

**Initial (0.33)**

*A punk rock frog in a studded leather jacket shouting into a microphone while standing on a boulder*

**Initial (0.00)**

*a cup on the right of a dog*

**Ours (1.00)**

*Hyperrealistic digital art: a fiercely passionate punk frog vocalist in a studded leather jacket, aggressively performing into a vintage microphone atop a colossal, weathered boulder, dramatic rim lighting, intricate textures, capturing raw energy and emotion.*

**Ours (1.00)**

*A friendly Golden Retriever gazes forward, a rustic stoneware mug sits to its right on a wooden surface.*

**Paraphrase (0.89)**

*Intricate illustration of a punk rock frog vocalist wearing a black studded leather jacket, passionately shouting into a vintage microphone while standing confidently on a large, weathered boulder, stage lights, high detail.*

**Paraphrase (0.00)**

*A dog and a cup, cup positioned right.*

**PAG (0.00)**

*hyper realistic, dramatic, intricate, elegant, highly detailed, digital painting, concept art, matte, sharp focus, illustration, art by wlop and greg rutkowski*

**PAG (0.00)**

*a cup on the right of a dog, hyperrealistic, hyper detailed, 8 k resolution, by wlop and greg rutkowski*

**Promptist (0.22)**

*a punkrock frog in a studded leather jacket shouting into a microphone while standing on a boulder,*

**Promptist (0.00)**

*A cup on the leftof a dog, a fantasy digital painting by Greg Rutkowski and James Gurney, trending on Artstation, highly detailed*

**OPT2I (0.33)**

*A screaming punk rock frog atop a boulder, microphone in hand, wearing a black leather jacket with studs.*

(a) DSG - PartiPrompt

(b) UniDet2D

Figure 11: Additional qualitative results for DSG (left) and UniDet2D (right),

**Initial (0.26)**

*a suitcase in front of a horse*

**Ours (0.66)**

*A majestic horse, dappled gray or chestnut, pauses before a vintage suitcase in a sunlit meadow, emphasizing photorealistic detail and soft, cinematic lighting.*

**Paraphrase (0.46)**

*A brown leather suitcase in a field, facing a stunning horse, 8k, photorealistic.*

**PAG (0.25)**

*a suitcase in front of a horse, hyper realistic, 8 k, by wlop and greg rutkowski*

**Promptist (0.53)**

*a suitcase in front of a horse, by greg rutkowski, digital art, realistic painting, fantasy, very detailed, trending on artstation*

**Initial (0.75)**

*one person and three cats*

**Ours (1.00)**

*A cinematic portrait of a person and three cats, a warm atmosphere, soft focus, and highly detailed textures, 8k resolution.*

**Paraphrase (0.75)**

*Intimate portrait: one person and three cats, natural light, detailed fur.*

**PAG (0.75)**

*a single person and three cats, hyperrealistic, hyperdetailed, 8 k realistic, by wlop and greg rutkowski*

**Promptist (0.50)**

*one person and three cats. intricate, elegant, highly detailed, digital painting, artstation, concept art, sharp focus, illustration, by justin gerard and artgerm, 8 k*

(a) UniDet3D                    (b) UniDetNumeracy

Figure 12: Additional qualitative results for UniDet3D (left) and UniDetNumeracy (right),

**Initial (90.00)**

*a hyperrealistic illustration of a monster covered in sand.*

**Ours (94.40)**

*Hyperrealistic: ancient, colossal monster *unfurling* from the desert, sand cascading, hyperdetailed scales, dramatic volumetric lighting, dust.*

**Paraphrase (92.60)**

*Hyperrealistic illustration of a monster completely covered in desert sand, intricate details.*

**PAG (90.00)**

*hyperrealistic illustration of a monster covered in sand, hyperdetailed, 8 k realistic, by wlop and greg rutkowski*

**Promptist (90.00)**

*a hyperrealistic illustration of a monster covered in sand, by greg rutkowski and thomas kinkade, trendingrete artstation.*

**Rule-Based (92.60)**

*a hyperrealistic illustration of a monster covered in sand, digital art, concept art, intricate*

**Initial (70.00)**

*a fantasy hybrid creature that has the body of a grizzly bear and the head of an owl*

**Ours (90.00)**

*Intricate fantasy hybrid: muscular grizzly bear and intelligent owl, seamless blend, dramatic god rays, trending on ArtStation.*

**Paraphrase (87.40)**

*Intricate and highly detailed fantasy creature – a grizzly bear's robust body elegantly combined with the head of a barn owl, atmospheric lighting, digital art.*

**PAG (49.00)**

*fantasy hybrid creature that looks like a grizzly bear and the head of an owl, hyper detailed, digital art, trending in artstation, cinematic lighting, by wlop*

**Promptist (71.00)**

*Fantasy hybrid creature that has the body of a grizzly bear and the head of an owl, by Greg Rutkowski and Raymond Swanland, Trending on Artstation, ultra realistic digital art*

**Rule-Based (48.00)**

*a fantasy hybrid creature that has the body of a grizzly bear and the head of an owl, elegant, intricate, fantasy.*

(a) Lexica - LLMGrader                          (b) DiffusionDB - LLMGrader

Figure 13: Additional qualitative results for LLMGrader in Lexica (left) and DiffusionDB (right) datasets.

---

**Initial Prompt**

An ultradetailed illustration of cthulu destroying a fleet of battleships                                (-0.044)

**Example Context for Hint-Generator LLM**

A stunning, ultradetailed illustration depicting Cthulhu rising from the abyssal depths to obliterate a fleet of battleships. Intense, dramatic lighting, Lovecraftian horror, colossal scale.                    (0.160)

Hyperdetailed Cthulhu, colossal tentacles crushing battleships in a bioluminescent abyss. Dramatic, volumetric lighting. Lovecraftian horror.                                                                (0.088)

**Generated Hint**

Focus on highly detailed, cinematic descriptions emphasizing scale, lighting, and Lovecraftian horror, while maintaining a balance between descriptive length and conciseness.

**Example Output from Optimizer LLM**

An intensely detailed, cinematic illustration of Cthulhu unleashing his wrath upon a fleet of battleships. Bioluminescent abyss, colossal scale, intricate tentacle details, dramatic, god-like lighting, pure Lovecraftian horror.                                                                                 (0.323)

An ultradetailed digital painting of Cthulhu's attack: immense tentacles rise from a bioluminescent abyss, destroying a fleet of battleships. Volumetric lighting, Lovecraftian horror aesthetic.            (0.273)

---

Figure 14: Additional case study of generated hints for Promptist Reward in Lexica dataset.

---

**Initial Prompt**

illustration of a colorful melting human head                                                          (-0.083)

**Example Context for Hint-Generator LLM**

Vibrant, dripping colors cascade over a melting human head – a highly detailed digital painting with flowing, bioluminescent textures – surreal and captivating.                                          (1.362)

A surreal and captivating illustration: a melting human head, rendered with hyperdetailed, dripping paint and bioluminescent textures. Dreamlike, liquid light.                                           (1.016)

**Generated Hint**

Combine strong keywords like "highly detailed," "surreal," "bioluminescent," and "dripping" with descriptions of flowing, organic textures and captivating ambiance, emphasizing it as an illustration or digital painting.

**Example Output from Optimizer LLM**

Captivating digital painting: a melting human head, enveloped in bioluminescent, dripping hues and flowing, organic textures – a surreal portrait with intricate details and a mesmerizing ambiance.   (1.390)

A highly detailed illustration of a melting human head, dripping with vibrant bioluminescent colors, showcasing flowing, organic forms and a surreal, captivating atmosphere.                         (0.903)

---

Figure 15: Additional case study of generated hints for Promptist Reward in DiffusionDB dataset.

---

**Initial Prompt**

Anthropomorphic mushroom.                                                                 (-0.380)

**Example Context for Hint-Generator LLM**

Full body portrait of a joyful anthropomorphic mushroom character, radiating positive energy, detailed clothing adorned with glowing moss and sparkling spores, lush magical forest, Artgerm and Ghibli style, 8k resolution, highly detailed.                                              (0.492)

8k full body: A benevolent anthropomorphic mushroom, radiating positive energy, wearing intricately woven clothing with glowing moss, lush magical forest environment, Artgerm style.        (0.748)

**Generated Hint**

Focus on "benevolent" or "joyful" anthropomorphic mushroom spirits/people with intricately woven, glowing moss attire in a lush, sun-dappled magical forest, specifying 8k full body portrait and Artgerm/Ghibli inspiration.

**Example Output from Optimizer LLM**

A joyful anthropomorphic mushroom character with a warm expression, detailed clothing of glowing moss and woven fibers, standing in a magical forest, 8k, Artgerm and Ghibli aesthetics.      (0.744)

Intricate 8k full body portrait of a benevolent mushroom person, radiating joy, with exquisitely woven attire of glowing moss, bathed in sunlight within a lush, magical forest – Artgerm & Ghibli inspired. (0.574)

---

Figure 16: Additional case study of generated hints for ImageReward in Lexica dataset.

---

**Initial Prompt**

a character wearing a white costume                                                       (-0.861)

**Example Context for Hint-Generator LLM**

Full body, white feathered costume, sculpted ice accents, WLOP style, volumetric light, detailed textures.                                                                               (0.110)

Full body illustration of a powerful character in a breathtaking white costume of sculpted ice, iridescent feathers, and shimmering silk, dramatic volumetric lighting.                 (0.744)

**Generated Hint**

Focus on detailed descriptions of materials (ice, feathers, silk) combined with strong lighting and artistic style references (WLOP, Greg Rutkowski, Artgerm) within a full-body illustration.

**Example Output from Optimizer LLM**

Full body illustration of a powerful character wearing a breathtaking white costume of sculpted ice, iridescent feathers, and shimmering frost, dynamic pose, dramatic volumetric lighting, Artgerm & Greg Rutkowski, intricate details, cinematic composition.                        (0.869)

Full body portrait: a character adorned in a white costume – ice, feathers, silk – dynamic pose, Artgerm & Greg Rutkowski, intricate details, rim lighting.                           (0.720)

---

Figure 17: Additional case study of generated hints for ImageReward in DiffusionDB dataset.

---

**Initial Prompt**

two baseballs to the left of three tennis balls (0.400)

**Example Context for Hint-Generator LLM**

Two baseballs, three tennis balls – arranged left to right. (0.667)

A close-up of two baseballs positioned to the left of three tennis balls, high detail, realistic materials, 8k, sharp focus, natural lighting. (0.333)

**Generated Hint**

Keep the prompt concise, focusing on quantity and left-to-right arrangement, and avoid overly descriptive or artistic terms

**Example Output from Optimizer LLM**

Two baseballs positioned left, followed by three tennis balls. (0.867)

Baseball x2, tennis balls x3, left to right. (0.667)

---

Figure 18: Additional case study of generated hints for DSG in PartiPrompt dataset.

---

**Initial Prompt**

a butterfly on the right of a balloon (0.000)

**Example Context for Hint-Generator LLM**

Swallowtail butterfly, right of a crimson balloon. (0.330)

Monarch butterfly, right of lavender balloon, detailed wings. (0.272)

**Generated Hint**

Focus on simpler prompts with key elements – butterfly type, balloon color, and "right of" – avoiding excessive detail or artistic jargon.

**Example Output from Optimizer LLM**

A blue Morpho butterfly to the right of a yellow balloon. (0.331)

Monarch butterfly right of red balloon. (0.329)

---

Figure 19: Additional case study of generated hints for UniDet2D.

---

**Initial Prompt**

a wallet in front of a person (0.060)

**Example Context for Hint-Generator LLM**

Realistic photo of a person with a wallet visibly held. (0.172)

Warm, candid portrait of a person, a wallet visible in foreground, shallow depth of field, photorealistic. (0.184)

**Generated Hint**

Focus on warm, candid portraits with the wallet visible in the foreground, emphasizing photorealism and shallow depth of field.

**Example Output from Optimizer LLM**

Photorealistic, warm-toned portrait, wallet visible, focused on hands, candid pose. (0.204)

Intimate, candid portrait of a person, wallet held naturally in foreground, soft, diffused light, high detail, photorealistic rendering, shallow depth of field. (0.102)

---

Figure 20: Additional case study of generated hints for UniDet3D.

**Initial Prompt**

three bicycles, two frogs, two keys, two helmets and two pigs                                                                (0.167)

**Example Context for Hint-Generator LLM**

A vibrant and detailed illustration: three bicycles, two emerald frogs, two golden keys, two scarlet helmets, and two rosy pigs – storybook aesthetic, trending on Artstation.                                                                (0.100)
Three bicycles with two frogs, keys, helmets, and pigs.                                                                (0.300)

**Generated Hint**

Focus on concise descriptions of the subjects and prioritize quantity of subjects over excessive detail or artistic style requests.

**Example Output from Optimizer LLM**

Two frogs, two keys, two helmets, two pigs, and three bicycles.                                                                (0.300)
A grouping of three bicycles, two frogs, two keys, two helmets, and two pigs.                                                                (0.267)

Figure 21: Additional case study of generated hints for UniDetNumeracy.

**Initial Prompt**

Painting of group of hungry alien snails homeless at dinner in a garbage dump.                                                                (53.00)

**Example Context for Hint-Generator LLM**

Highly detailed painting of forlorn alien snails sharing a meager meal within a realistically decaying garbage dump, volumetric lighting, melancholic mood.                                                                (79.53)
Highly detailed digital painting of desperately hungry, homeless alien snails gathered for a somber dinner amongst the refuse of a vast, decaying garbage dump.                                                                (66.33)

**Generated Hint**

Focus on combining "highly detailed," "dramatic," "realistic rendering/decaying," and "volumetric/rim lighting" with clear emotional descriptors like "forlorn," "desperate," or "melancholic" while specifying "dinner" or "meal" within the "garbage dump" setting.

**Example Output from Optimizer LLM**

Highly detailed, atmospheric painting: desperately hungry, homeless alien snails gathered for a melancholic dinner amongst the refuse of a sprawling, realistically decaying garbage dump, dramatic lighting.                                                                (86.73)
Realistic rendering of forlorn alien snails sharing a sparse meal in a sprawling garbage dump, highly detailed textures, dramatic rim lighting, melancholic mood.                                                                (76.13)

Figure 22: Additional case study of generated hints for LLMGrader Reward in Lexica Dataset.

**Initial Prompt**

a phoenix                                                                (91.33)

**Example Context for Hint-Generator LLM**

A breathtaking phoenix, intensely detailed plumage, swirling flames, digital painting, art by Artgerm. (93.67)
A phoenix reborn from ashes, highly detailed plumage, dramatic lighting, digital painting, Artgerm inspired.                                                                (93.13)

**Generated Hint**

Combine strong artistic references (Artgerm & Rutkowski are consistently helpful), emphasize intricate detail in plumage/feathers, and use evocative descriptions of fire/light (molten gold, god rays, swirling embers) with dramatic lighting and epic scale.

**Example Output from Optimizer LLM**

Majestic phoenix with feathers of shimmering gold and crimson, dynamic pose, dramatic lighting, Rutkowski, swirling embers, highly detailed fantasy art.                                                                (94.33)
A breathtaking phoenix, intensely detailed iridescent plumage, engulfed in swirling flames, digital painting, dramatic lighting.                                                                (94.00)

Figure 23: Additional case study of generated hints for LLMGrader Reward in DiffusionDB dataset.

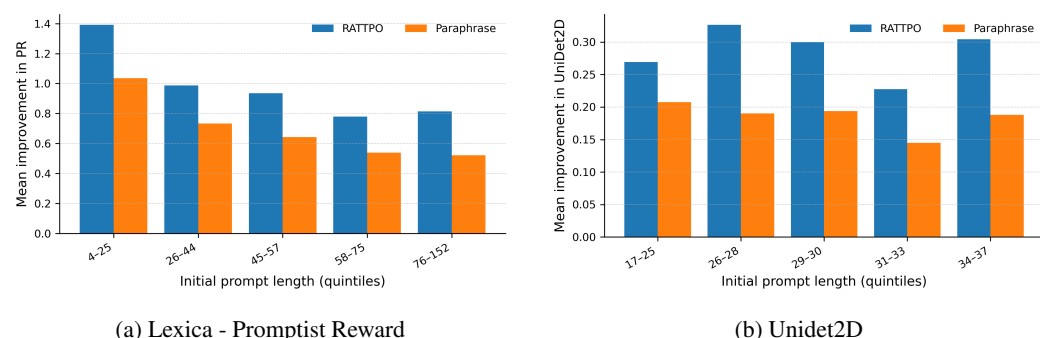

(a) Lexica - Promptist Reward                    (b) Unidet2D

Figure 24: Average reward gain over the initial prompt, grouped by initial prompt length.

Table 16: Ablation study with different LLMs (GPT family).

| Method | PR | UniDet2D |
|---|---|---|
| Initial | -0.325 | 0.133 |
| RATTPO + Gemma 3 27B | 0.683 | 0.416 |
| RATTPO + GPT 4.1 mini | 0.418 | 0.432 |
| RATTPO + GPT 4o mini | 0.405 | 0.414 |

## J  ADDITIONAL ANALYSIS AND ABLATION STUDY

### J.1  IMPACT OF INITIAL PROMPT LENGTH ON OPTIMIZATION

To identify how the initial prompt length affects the effectiveness of the prompt optimization. We group initial prompts into length quintiles and, for each bin, report the average absolute reward gain over the initial prompt. The experimental setup follows our ablation studies, and the results are shown in Fig. 24. Across all length bins, RATTPO consistently achieves substantial reward improvements, and its gain is larger than that of the Paraphrase baseline in every bin, indicating that RATTPO effectively optimizes both short and long prompts. In the PR+Lexica setting, we further observe that shorter prompts benefit particularly strongly from optimization, which is intuitive since adding detailed scene, object, or style descriptions can substantially improve the aesthetics of generated images. In contrast, under the UniDet2D reward, no such correlation appears, likely because spatial composition scores do not necessarily benefit from longer or more descriptive prompts. These patterns illustrate that different reward models exhibit quite different prompt preferences. Despite this diversity, RATTPO adapts reliably to each underlying reward via the hint mechanism, yielding consistent, substantial gains across all bins.

### J.2  RATTPO WITH DIFFERENT LLM FAMILY

To see how the optimization performance changes with the choice of different LLM family, we conduct an experiment using different LLM family. Specifically, we evaluate RATTPO using two GPT-family models (`gpt-4.1-mini-2025-04-14` and `gpt-4o-mini-2024-07-18`), and the results are in Tab. 16. Across both backbones, RATTPO consistently improves the target reward compared to the initial prompt. While the magnitude of improvement varies, which we attribute to differences in paraphrasing behavior and in-context learning ability of each backbone, the overall trend shows that RATTPO can be reliably applied to diverse LLMs.

### J.3  MERGED SINGLE-LLM ABLATION

To identify whether the dual-LLM loop is essential, we conduct an ablation study with single-LLM variant. The Single-LLM variant is prompted to output both the rewritten prompt and the hint about the underlying reward in one loop, instead of using separate optimizer and hint-generator LLMs. The results are shown in Tab. 17. While the single-LLM variant improves over the initial prompt, it consistently underperforms the dual-LLM RATTPO design. These results indicate that separating

Table 17: Ablation study: dual-LLM

| Method | PR | UniDet2D |
|---|---|---|
| Initial | -0.325 | 0.133 |
| RATTPO (dual-LLM loop) | **0.683** | **0.416** |
| Single-LLM | 0.482 | 0.389 |

Table 18: Cross-model transfer of RATTPO-optimized prompts (PR + Lexica).

| Method | SD1.4 → SD2.1 | SD1.4 → SDXL-Turbo |
|---|---|---|
| Initial | -0.373 | -0.327 |
| Paraphrase | -0.210 | -0.215 |
| RATTPO | -0.140 | -0.180 |

Table 19: Cross-model transfer of RATTPO-optimized prompts (UniDet2D).

| Method | SD1.4 → SD2.1 | SD1.4 → SDXL-Turbo |
|---|---|---|
| Initial | 0.133 | 0.162 |
| Paraphrase | 0.156 | 0.201 |
| RATTPO | 0.184 | 0.201 |

the roles of optimization and hint generation is beneficial, and that the dual-LLM architecture is an important component of RATTPO. Intuitively, decoupling these roles simplifies the objective of each LLM, whereas asking a single LLM to jointly optimize prompts and infer general reward hints appears to be a more challenging problem.

### J.4 TRANSFERABILITY

We conduct an experiment to identify the transferability of the optimized prompts to other T2I models. We consider T2I models with different text encoders: SD1.4 (CLIP ViT-L), SD2.1 (OpenCLIP ViT-H), and SDXL Turbo (OpenCLIP ViT-bigG with CLIP ViT-L), and measure the reward of the transferred prompts. As shown in Tab. 18 and Tab. 19, RATTPO-optimized prompts exhibit positive transfer, consistently showing improved reward upon initial prompt. However, the gain decreases as the models differ more, for example when transferring from SD1.4 to SDXL-Turbo compared to SD1.4 to SD2.1. For practical use, we therefore recommend running RATTPO directly on the target T2I model, especially since our experiments show that RATTPO works robustly across diverse T2I backbones (Tab. 6).

## K EXTENDED COMPARISON WITH BASELINES

For a more complete comparison, we additionally compare our method with two baselines (Mañas et al., 2024; Liu et al., 2024).

**Comparison with Mañas et al. (2024)** We evaluate RATTPO under all four experimental setups used in OPT2I (Mañas et al., 2024) and summarize the results in Fig. 25. When implementing OPT2I, we used the same metaprompts as in their paper, which were separately designed for DSG and dCS.

Across all these settings, RATTPO consistently outperforms OPT2I except in DSG+COCO. In DSG+COCO, the performance gap is naturally small because the initial prompts already achieve high DSG scores (around 40% are perfect and the reward is upper bounded by 1), which leaves limited headroom for improvement. The fact that RATTPO matches or surpasses OPT2I under OPT2I's own settings also suggests an interesting observation: manually engineered reward structures and metaprompts in OPT2I can be suboptimal, and that a reward-agnostic adaptation strategy like RATTPO can better optimize the target reward.

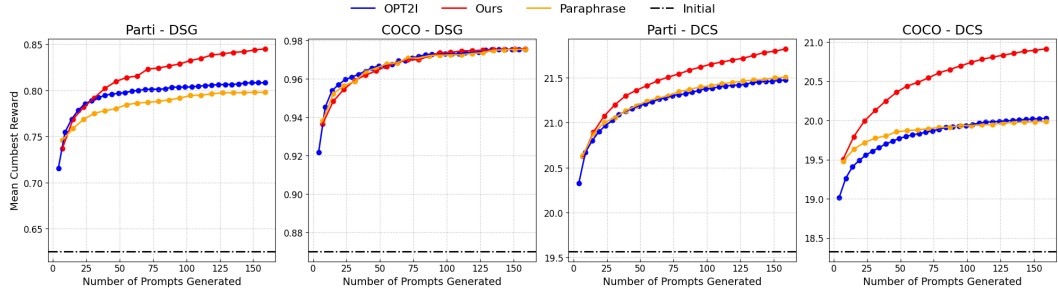

Figure 25: Additional experiment under all experiment settings employed in OPT2I.

Table 20: Comparison with Liu et al. (2024) under the ablation experiment setting.

| Method | PR | UniDet2D |
|---|---|---|
| Initial | -0.325 | 0.133 |
| Liu et al. (2024) | -0.061 | 0.094 |
| Paraphrase | 0.372 | 0.316 |
| RATTPO (dual-LLM loop) | **0.683** | **0.416** |

**Comparison with Liu et al. (2024)** We implement the prompt optimization framework described in Section 6 of (Liu et al., 2024) and evaluate it under our ablation experiment setting. The official meta-prompt from their released code is used, and we match the backbone LLM and prompt budget across all methods. The results are reported in Tab. 20.

As can be seen, Liu et al. (2024) performs noticeably worse than RATTPO and even underperforms the simple Paraphrase baseline. We believe this gap stems from structural differences between the methods. In (Liu et al., 2024), a single multimodal LLM is used both to critique the image and to rewrite the prompt, so the optimization is implicitly tied to that model's internal judgment and cannot adapt to arbitrary external rewards. Moreover, the feedback at each step is based only on the current prompt and last generated image, without explicitly leveraging the full optimization history. In contrast, RATTPO can effectively adapt to external specified reward at test time, and its dual-LLM loop is explicitly designed to summarize and exploit the optimization trajectory to aid searching for high-reward prompts.

## L    VISUAL ILLUSTRATION OF RATTPO

Fig. 26 shows a diagram that illustrates the full workflow of RATTPO, with important details about metaprompts.

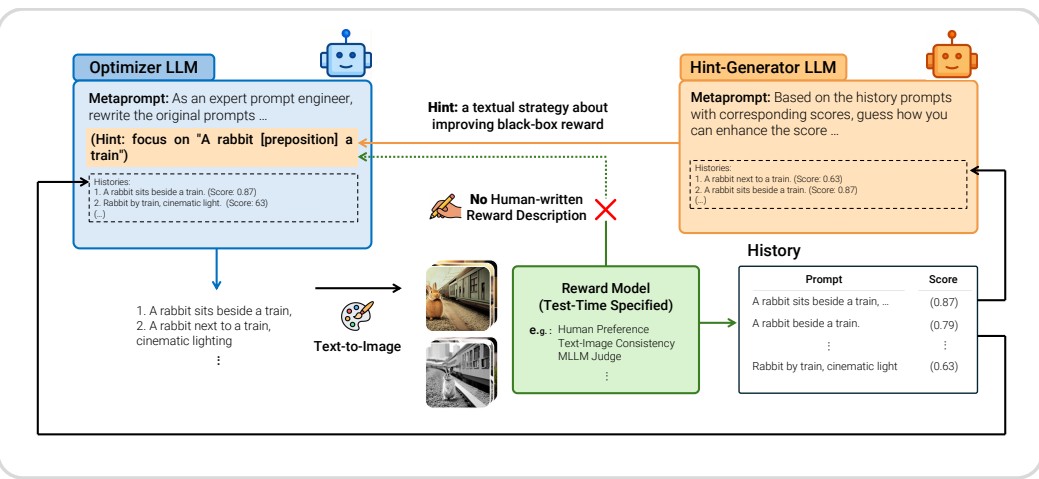

Figure 26: Overview of RATTPO. The dual-LLM optimization loop iteratively refines prompts using trajectory-based hints, enabling reward-agnostic test-time prompt optimization under test-time specified external reward models.

