# OpenReview forum: "Reward-Agnostic Prompt Optimization for Text-to-Image Diffusion Models"
_ICLR.cc/2026/Conference — Submitted to ICLR 2026_

### Official Review · Reviewer_EEck · 2025-10-27

**Soundness:** 3
**Presentation:** 3
**Contribution:** 2
**Rating:** 6
**Confidence:** 4

**Summary:**

This paper presents a well-motivated approach (RATTPO) for reward-agnostic prompt optimization for text-to-image generation. It iteratively refines an initial prompt by querying LLMs at test time: one optimizer LLM proposes new prompts conditioned on the optimization history and a hint-generator LLM to provide reward-aware feedback. Empirical results demonstrate the versatility and effectiveness of PATTPO across a wide range of rewards, including human preference, text-to-image consistency, and holistic MLLM assessment. RATTPO also shows higher search efficiency compared to other test-time search baselines.

**Strengths:**

1. The primary strength is the reward-agnostic nature of RATTPO, which is convincingly demonstrated across different diverse reward functions.

2. The method is training-free and gradient-free and exhibits superior generalization when compared to learning-based baselines.

3. The hint is formatted as natural language feedback, making the optimization process transparent and potentially human-interpretable.

**Weaknesses:**

1. Lack the ablation of using single prompting loop for both prompt generation and hint generation.
2. The method is computationally demanding, requiring two LLMs (optimizer and hint generator) in an iterative loop and necessitating multiple costly image generation and reward function calls (up to 160 generated prompts) to achieve good performance.

**Questions:**

1. Have you encountered the reward hacking problem in your optimization framework?
2. Why do you not consider integrating both optimizer and hint-generator in a single loop?

---

> ### Author Response · Authors · 2025-11-21
> **Rebuttal Response**
>
> We appreciate the reviewer's effort in providing feedback that helps us improve our work. We address the questions and weaknesses as below.
>
> ### **\[W1 / Q2\] Missing ablation of using single LLM loop.**
>
> We thank the reviewer for the helpful suggestion. To directly address it, we conducted an ablation where a single LLM is prompted to output both the rewritten prompt and the hint about the underlying reward in one loop, instead of using separate optimizer and hint-generator LLMs. The results are reported in Table A.
>
> **Table A. Ablation study: dual-LLM**
>
> |  | PR \+ Lexica | UniDet2D |
> | :---- | :---- | :---- |
> | Initial | \-0.325 | 0.133 |
> | dual-LLM (RATTPO) | 0.683 | 0.416 |
> | single-LLM | 0.482 | 0.389 |
>
> While the single-LLM variant improves over the initial prompt, it consistently underperforms the dual-LLM RATTPO design. These results indicate that separating the roles of optimization and hint generation is beneficial, and that the dual-LLM architecture is an important component of RATTPO. Intuitively, decoupling these roles simplifies the objective of each LLM, whereas asking a single LLM to jointly optimize prompts and infer general reward hints appears to be a more challenging problem. We have included this ablation in the revised version of the paper (Appendix J.3, Table 17).
>
> ### **\[W2\] Computational demand.**
>
> We agree with the reviewer that LLM calls, image generations, and reward evaluations introduce computational overhead. However, RATTPO is intentionally designed as a test-time optimization method that uses test-time compute to avoid the much larger training cost (often several GPU-days) required by learning-based prompt engineers. In many practical scenarios where reward models change or need to be personalized, this upfront training cost dominates, whereas RATTPO only relies on inference-time calls and can be directly applied to new, unseen rewards.
>
> Within this test-time regime, the computational cost of RATTPO is both moderate and controllable. As discussed in our search-efficiency analysis (Table 3), RATTPO achieves a similar reward to the reward-agnostic Paraphrase baseline while being on average 4.8 times faster in end-to-end runtime. Since already high rewards are often reached earlier than the budget of 160 prompts, one can stop once the reward plateaus. Moreover, the number of iterations and candidates per iteration can be reduced to match a given compute budget, and the image generation and reward evaluations for different candidates are independent and can be run in parallel, which helps keep wall-clock latency manageable.
>
> ### **\[Q1\] Reward hacking problem.**
>
> Reward hacking is an important concern for test-time optimization methods. To directly assess this, we conducted a cross-reward evaluation (Table 4), where we optimize prompts for one reward and evaluate them on another, largely uncorrelated reward. For completeness, we included the results again as below.
>
> **Table B: Cross-reward experiment results. Italicized numbers indicate that RATTPO was not optimized for the evaluated reward.**
>
> |  | PR (aesthetic reward) | UniDet2D (spatial reward) |
> | :---- | :---- | :---- |
> | Initial | \-0.311 | 0.159 |
> | Ours, optimizing PR | 1.021 | *0.164* |
> | Ours, optimizing UniDet2D | *\-0.017* | 0.461 |
>
> In both directions, the non-target reward stays stable or improves rather than collapsing, which suggests that RATTPO is not merely exploiting idiosyncrasies of a single reward at the expense of overall image quality.
>
> In addition, qualitative inspection of high-scoring prompts and images did not reveal adversarial or clearly “hacked” behaviors such as unnatural prompts or images that violate user intent. Most improvements come from adding fine-grained descriptive details and clarifying spatial relations. We hypothesize that operating in discrete text space, together with explicitly instructing the LLM to preserve the semantics of the initial prompt, acts as a regularizer that helps mitigate reward hacking.

---

> ### Author Response · Authors · 2025-11-28
> **Kind Reminder Regarding Rebuttal**
>
> Dear Reviewer EEck,
>
> We hope this message finds you well. We would like to kindly follow up regarding the rebuttal feedback for our submission. We truly appreciate the time and effort you have devoted to reviewing our paper, and we are happy to provide any clarifications if needed.
>
> We understand that this is a busy period and greatly value your consideration. Please let us know if any further information would be helpful from our side.
>
> With kind regards, \
> Authors of paper 11890

---

### Official Review · Reviewer_oSLi · 2025-10-28

**Soundness:** 2
**Presentation:** 2
**Contribution:** 3
**Rating:** 4
**Confidence:** 2

**Summary:**

This paper presents RATTPO, a reward-agnostic test-time prompt optimization method for text-to-image diffusion models. Unlike previous approaches that are tailored to specific reward functions, RATTPO can flexibly enhance prompts across various evaluation scenarios by leveraging LLMs and a novel reward-aware feedback signal. Experimental results show that RATTPO significantly improves search efficiency and prompt quality for diverse reward models. With adequate inference budget, RATTPO achieves performance comparable to specialized learning-based baselines without requiring task-specific tuning.

**Strengths:**

- The motivation and significance of the proposed scenario are clearly articulated and highly relevant.
- The experimental results convincingly demonstrate the superiority of the proposed method over existing approaches.

**Weaknesses:**

- The paper is poorly written, with an overly brief description of the methodology. It lacks essential details about the input prompts used for the first LLM to generate candidate prompts for image generation, the input prompts for the second LLM, and the specific format of the "hint" texts, all of which are critical to understanding the core approach.
- The paper lacks a clear diagram illustrating the overall workflow of the proposed method; Algorithm 1 alone is insufficient for conveying the process.
- Lines 054-057 contain two sentences that redundantly express the same idea.

**Questions:**

See Weaknesses.

---

> ### Author Response · Authors · 2025-11-21
> **Rebuttal Response**
>
> We appreciate the reviewer's effort in providing feedback that helps us improve our work. We address the questions and weaknesses as below.
>
> ### **\[W1\]\[W2\] Missing essential details. / Missing diagram for overall workflow**
>
> We kindly note that the current manuscript already includes the full meta-prompts for both LLMs in Appendix B, as well as concrete examples of the generated hints in our case study (Figure 4 in the main paper and Figures 14-23 in the appendix).
> For a clearer presentation, we added an overview diagram that illustrates the full workflow of RATTPO, which also includes some important details about metaprompts (Appendix L, Figure 26). We hope this revision clarifies the methodology, and we are happy to further refine the presentation if needed.
>
>
> ### **\[W3\] Redundant sentences in L054-L057.**
>
> We appreciate the reviewer for pointing out the redundancy. In the revised manuscript, we have merged them into a single, more concise sentence:
>
> - To address this challenge, we introduce Reward-Agnostic Test-Time Prompt Optimization (RATTPO), a flexible test-time optimization approach for automated prompt engineering under diverse reward functions that are only specified at test time.

---

> ### Author Response · Authors · 2025-11-28
> **Kind Reminder Regarding Rebuttal**
>
> Dear Reviewer oSLi,
>
> We hope this message finds you well. We would like to kindly follow up regarding the rebuttal feedback for our submission. We truly appreciate the time and effort you have devoted to reviewing our paper, and we are happy to provide any clarifications if needed.
>
> We understand that this is a busy period and greatly value your consideration. Please let us know if any further information would be helpful from our side.
>
> With kind regards, \
> Authors of paper 11890

---

> > ### Comment · Reviewer_oSLi · 2025-11-28
> >
> > Thank you for your responses to my review.
> >
> > My main concern pertains to the quality of the paper’s writing. While the author has addressed my concerns to some extent, I still believe that the writing could be further improved.
> >
> > However, I would like to note that my expertise does not lie in the primary domain of this paper, as reflected in my confidence score of 2. I also observed that the first two reviewers, who seem to be more familiar with the technical aspects, have raised similar concerns and given the same score as mine.
> >
> > Therefore, I will maintain my current score for now. Should the first two reviewers revise their scores, I am willing to follow their decision.

---

### Official Review · Reviewer_gfqR · 2025-10-31

**Soundness:** 2
**Presentation:** 3
**Contribution:** 1
**Rating:** 4
**Confidence:** 4

**Summary:**

In this paper, the authors propose a new method that uses LLMs to perform automated prompt engineering for text-to-image generation. In particular, they suggest using an LLM as “prompt optimizer” and another LLM as “hint generator” to iteratively improve the generated prompts based on images generated from a text-to-image generative model and an external reward model. At every iteration, the LLM prompt optimizer will generate a prompt given the history of improvements, then the text-to-image model will generate images based on the new prompt and the reward model will output the reward w.r.t. the generated images. The LLM hint generator then produces edit suggestions for the prompt and then the LLM prompt optimizer will improve the prompts produced based on these suggestions. They also conduct experiments to show the effectiveness of their method in comparison to multiple baselines on various datasets.

**Strengths:**

1. The experimental results look very promising, especially in Figure 1 where they show great test time scaling.
2. The algorithm is fairly simple and easy to implement.
3. The paper is well written and easy to understand.

**Weaknesses:**

1. My main concern about the paper is regarding its novelty. The idea of both LLM as automated prompt generator and as a judge/hint giver has been thoroughly explored both in the context of LLM self-improvement/RLAIF [2,3,4] [(Madaan et al., 2023; Wang et al.,
2023a; Shinn et al., 2023) from the paper] and text-to-image generation [1] [(Yang et al.,
2024; Fernando et al., 2023; Du et al., 2024; He et al., 2024; Mañas et al., 2024) from the paper]. In fact, the algorithm proposed in this paper is strikingly similar to [1] and He et al. 2024. It is unclear to me the marginal changes made in this paper are significant enough.
2. The experiment results, while showing a lot of promise, do seem a bit selective and incomplete. For example,

    (i) In Figure 1, OPT2I only shows up in one out of eight subplots, which is also the only place where this baseline is compared. Given the extreme similarity of the methodology, it would make sense for the authors to include OPT2I in all comparisons that they conduct. Similarly, somehow not all baselines are compared in all experiments.

    (ii) When comparing the inference time, the authors denote the wallclock time for their method as “Time, RATTPO at win”, which seems to indicate that they are only accounting for the cases where their method outperformed the baseline. It is very unclear why they would make this selection, i.e. why not just calculate the wallclock time for all RATTPO runs?
3. Besides the concerns above, the authors should also consider adding the following experiments to strengthen the paper:

	(i) The authors should include the comparison against [1], as it is a highly related and similar work (specifically section 6 in [1])

	(ii) The authors only use the Gemma model family in their experiment and they should consider other MLLMs like the GPT family, etc.


Reference:

[1] Liu et al. Language Models as Black-Box Optimizers for Vision-Language Models. 2024.

[2] Chao et al. Jailbreaking black box large language models in twenty queries. 2023.

[3] Wang et al. Self-Instruct: Aligning Language Models with Self-Generated Instructions. 2022.

[4] Huang et al. Large Language Models Can Self-Improve. 2022.

[5] Lee et al. RLAIF vs. RLHF: Scaling Reinforcement Learning from Human Feedback with AI Feedback. 2024.

**Questions:**

How transferable are the prompts that are optimized for one text-to-image model to another one?

---

> ### Author Response · Authors · 2025-11-21
> **Rebuttal Response [1/3]**
>
> We appreciate the reviewer's effort in providing feedback that helps us improve our work. We address the questions and weaknesses as below.
>
> ### **[W1] Novelty**
>
> We would like to clarify the contribution of RATTPO, and how RATTPO differs from prior "LLM-as-optimizer" loops and multi-LLM systems. Our main contribution is proposing an  automated T2I prompt engineer that can adapt to **unseen** reward at test-time, which shapes unique design choices and distinct role to the second LLM:
>
> - **Adaptation to reward specified at test time**. To our knowledge, RATTPO is the first reward-agnostic prompt engineer designed to adapt to the target reward provided at test time, without reward-specific metaprompts or task descriptions. Prior LLM-as-optimizer approaches for T2I prompt engineering were developed under a known-reward assumption, which limits applicability when evaluation rubrics vary in practice.
> - **Unique role of the hint-generator LLM**. The second LLM in RATTPO infers a general, reward-aware strategy ("hint") from observed (prompt, score) trajectories and conditions the optimizer LLM. The role is to analyze the characteristics of the unknown reward, and is considerably different to previous works that mainly focus on judging / scoring generated output.
>
> We elaborate with comparisons to the prior works below.
>
> **Reward-Agnostic T2I Prompt Engineer** Our goal is a reward-agnostic automated T2I prompt engineer that adapts across heterogeneous rewards plugged in at test time. This is valuable in real deployments where evaluators and rubrics evolve. In contrast, earlier LLM-as-optimizer methods for prompt engineering bind the optimizer to reward-specific instructions or setups, inducing another form of human prompt engineering. Simply discarding such descriptions is also suboptimal; in RATTPO, on-the-fly hints provide the needed guidance without hard-coding reward details (Tab. 2).
>
> Concretely, prior systems are reward-specific:
> - He et al. [1] target personalized generation and vary prompting by task (e.g., subject-driven, style-driven, inversion), assuming a known objective per task.
> - OPT2I [2] aims at T2I consistency with reward-specific metaprompts tailored to particular consistency scorers; generalization to unseen rewards is not the design focus.
> - Liu et al. [3] optimize under fixed, task-specific objectives/metrics; they do not treat the reward model itself as a variable specified at test time.
>
> In contrast, RATTPO uses one reward-agnostic optimizer prompt across diverse, unseen, heterogeneous rewards (human preference, compositionality, holistic MLLM grading) and multiple diffusion backbones, without redesigning metaprompts.
>
> **Distinct Role of the Hint-Generator LLM** Since we target unknown-reward optimization, we need an automated mechanism to replace human-written reward descriptions in the optimizer's metaprompt. In RATTPO, the second LLM plays this role: it takes sampled (prompt, reward) histories and infers a succinct, reward-aware strategy that guides the optimizer LLM. This is materially different from prior dual-LLM or judge-in-the-loop designs:
>
> - Within T2I, He et al. [1] is the closest dual-LLM loop we are aware of, but the second LLM is positioned primarily as an evaluator/judge under a known objective. In RATTPO, the second LLM generates strategy text while scoring remains external; this enables adaptation to arbitrary black-box rewards specified at test time.
> - Outside T2I, many multi-LLM/self-improvement works use LLMs as judges/critics under fixed tasks: e.g., Chao et al. [4] use an LLM judge for jailbreak success; Self-Refine [5] employs self-critique based on internal criteria; Reflexion [6] stores reflections under task-specific rewards. These systems do not treat the reward model as an externally changeable object at test time.
>
> In contrast, RATTPO's dual-LLM design is structurally different: the second LLM is not a scorer but a reward-surrogate hint generator that replaces hand-crafted reward descriptions and enables a single optimizer LLM to adapt across heterogeneous rewards. As revealed in our case study in Figure 4 and Figures 14-23, the hint-generator LLM can indeed identify the optimization strategy to improve underlying black-box reward, without any reward-specific modification.
>
> Taken together, we believe that RATTPO represents a significant step toward fully automated, versatile prompt engineering that functions in real-world scenarios.
>
> [1] He et al. Automated black-box prompt engineering for personalized text-to-image generation. 2025\
> [2] Mañas et al. Improving text-to-image consistency via automatic prompt optimization. 2025\
> [3] Liu et al. Language Models as Black-Box Optimizers for Vision-Language Models. 2024\
> [4] Chao et al. Jailbreaking black box large language models in twenty queries. 2023\
> [5] Madaan et al. Self-Refine: Iterative Refinement with Self-Feedback. 2023\
> [6] Shinn et al. Reflexion: language agents with verbal reinforcement learning. 2023

---

> ### Author Response · Authors · 2025-11-21
> **Rebuttal Response [2/3]**
>
> ### **[W2(i)] Baseline comparisons**
>
> Since our goal is to demonstrate the generality of RATTPO for diverse unseen reward functions, we choose eight rewards to cover a broad spectrum of reward types (differentiable vs. non-differentiable, aesthetics vs. object relations, and rewards from human-preference models, detectors, VQA, and MLLMs). Since most prior works employ the task-specific designs (e.g., meta-prompts, critiques, etc.), we could not apply these methods on datasets where such designs are not available. Under this setting, the experiment overlaps with OPT2I only in the DSG+Parti configuration, which is why OPT2I was originally included there.
> To address the reviewer’s concern and provide a more complete comparison, we additionally evaluate RATTPO under all four experimental setups used in OPT2I and summarize the results in Table A (with full optimization curves in the Appendix K, Figure 25). Note that OPT2I employs specialized meta-prompts for each reward setups (DSG and dCS), while our method is consistently applied across rewards without any manual tuning.
>
> **Table A: Additional experiment under all experiment settings employed in OPT2I. Optimization curves are shown at Figure 25.**
>
> |  | DSG+Parti | DSG+COCO | dCS+Parti | dCS+COCO |
> | - | - | - | - | - |
> | Initial | 0.625 | 0.870 | 19.57 | 18.33 |
> | Paraphrase | 0.791 (+0.166) | 0.975 (+0.105) | 21.50 (+1.93) | 19.99 (+1.66) |
> | OPT2I | 0.802 (+0.177) | 0.975 (+0.105) | 21.47 (+1.90) | 20.03 (+1.70) |
> | RATTPO | **0.842 (+0.217)** | **0.976 (+0.106)** | **21.82 (+2.25)** | **20.91 (+2.58)** |
>
> Across all these settings, RATTPO consistently outperforms OPT2I except in DSG+COCO. In DSG+COCO, the performance gap is naturally small because the initial prompts already achieve high DSG scores (around 40% are perfect and the reward is upper bounded by 1), which leaves limited headroom for improvement. The fact that RATTPO matches or surpasses OPT2I under OPT2I’s own settings also suggests an interesting observation: manually engineered reward structures and metaprompts in OPT2I can be suboptimal, and that a reward-agnostic adaptation strategy like RATTPO can better optimize the target reward.
>
> For other baselines, not all baselines are included in all setups to avoid misleading or unavailable comparison, as explained in the Appendix A. For instance, DPO-Diff's gradient-based optimization cannot be applied when the reward is non-differentiable. Also, the rule-based baseline is built from empirically tuned heuristics for image aesthetics, and similarly well-established heuristic rules for text-image consistency are, to our knowledge, not available.
>
> ### **\[W2(ii)\] Clarification on search efficiency calculation**
>
> We would like to clarify how the search efficiency in Table 3 and Table 8 is computed. The reported “Time, RATTPO at win” does not select only successful runs; it is based on averages over all RATTPO runs.
> Specifically, we first record the average reward and wall-clock time of the Paraphrase baseline at a fixed prompt budget of 160\. For RATTPO, we then track the average reward and cumulative runtime at each search round, again averaged over all runs. We identify the earliest search round at which RATTPO’s average reward meets or exceeds the Paraphrase baseline’s average reward, and the corresponding prompt budget is reported (Average \# of prompts at win). The associated runtime (Time, RATTPO at win) is used to calculate the speedup in the table, which is the ratio between the baseline runtime and this RATTPO runtime. With this calculation, the metric measures the average time required for RATTPO to reach the same reward level as the baseline, rather than filtering for only those instances where RATTPO happens to outperform it. We have revised the manuscript to explicitly describe this procedure and avoid confusion.

---

> ### Author Response · Authors · 2025-11-21
> **Rebuttal Response [3/3]**
>
> ### **\[W3(i)\] Comparison to (Liu et al.), section 6\.**
>
> As suggested by the reviewer, we implemented the prompt optimization framework described in Section 6 of Liu et al. and evaluated it under our ablation setting. We used the official meta-prompt from their released code and matched the backbone LLM and prompt budget across all methods. The results are reported in Table B below.
>
> **Table B: Comparison with Liu et al. under the ablation experiment setting. Same to Table 20 in Appendix K.**
>
> | Method | PR \+ Lexica | UniDet2D |
> | ----- | ----- | ----- |
> | Initial | \-0.325 | 0.133 |
> | Liu et al. | \-0.0607 | 0.0935 |
> | Paraphrase | 0.372 | 0.316 |
> | RATTPO | 0.683 | 0.416 |
>
> From Table B, Liu et al. performs noticeably worse than RATTPO and even underperforms the simple Paraphrase baseline. We believe this gap stems from structural differences between the methods. In Liu et al. \[1\], a single multimodal LLM is used both to critique the image and to rewrite the prompt, so the optimization is implicitly tied to that model’s internal judgment and cannot adapt to arbitrary external rewards. Moreover, the feedback at each step is based only on the current prompt and last generated image, without explicitly leveraging the full optimization history. In contrast, RATTPO can effectively adapt to external specified reward at test time, and its dual-LLM loop is explicitly designed to summarize and exploit the optimization trajectory to aid searching for high-reward prompts.
>
> ### **\[W3(ii)\] LLMs other than Gemma, e.g. GPT family.**
>
> As suggested by the reviewer, we conducted additional experiments with a different LLM family. Specifically, we evaluated RATTPO using two GPT-family models (gpt-4.1-mini-2025-04-14 and gpt-4o-mini-2024-07-18), and the results are in Table C. Across both backbones, RATTPO consistently improves the target reward compared to the initial prompt. While the magnitude of improvement varies, which we attribute to differences in paraphrasing behavior and in-context learning ability of each backbone, the overall trend shows that RATTPO can be reliably applied to diverse LLMs.
>
> **Table C: Ablation study with different backbone LLMs (GPT family). Same to Table 16 in Appendix J.2.**
> |  | PR \+ Lexica | UniDet2D |
> | :---- | :---- | :---- |
> | Initial | \-0.325 | 0.133 |
> | RATTPO \+ Gemma 3 27B | 0.683 | 0.416 |
> | RATTPO \+ GPT 4.1 mini | 0.418 | 0.432 |
> | RATTPO \+ GPT 4o mini | 0.405 | 0.414 |
>
> ### **\[Q1\] Transferability of optimized prompts**
>
> Regarding the transferability of optimized prompts across T2I models, we would like to first clarify our focus. Learning-based prompt engineers, in particular, must carefully consider cross-model transfer, since retraining for each new T2I backbone is costly (e.g., several GPU-days). In contrast, RATTPO is a test-time optimization method that can be directly applied to any target T2I model with modest compute. In practice, this makes it natural to run RATTPO on the target model rather than reusing the optimized prompt across different models.
>
> To directly address the reviewer’s question, we additionally evaluate how prompts optimized on one T2I model transfer to another. We consider T2I models with different text encoders: SD1.4 (CLIP ViT-L), SD2.1 (OpenCLIP ViT-H), and SDXL Turbo (OpenCLIP ViT-bigG with CLIP ViT-L), and measure the reward of the transferred prompts. The results are shown in Table D and Table E.
>
> **Table D. Cross-model transfer of RATTPO-optimized prompts (PR \+ Lexica). Same to Table 18 in Appendix J.4.**
> | PR, Lexica | SD1.4 → SD2.1 | SD1.4 → SDXL Turbo |
> | :---- | :---- | :---- |
> | (Initial) | \-0.373 | \-0.327 |
> | Paraphrase | \-0.210 | \-0.215 |
> | RATTPO | \-0.140 | \-0.180 |
>
> **Table E. Cross-model transfer of RATTPO-optimized prompts (UniDet 2D). Same to Table 19 in Appendix J.4.**
> | UniDet2D | SD1.4 → SD2.1 | SD1.4 → SDXL Turbo |
> | :---- | :---- | :---- |
> | (Initial) | 0.133 | 0.162 |
> | Paraphrase | 0.156 | 0.201 |
> | RATTPO | 0.184 | 0.201 |
>
> The results show that, although cross-model transfer is not the scope of our method, RATTPO-optimized prompts exhibit positive transfer, consistently showing improved reward upon initial prompt. However, the gain decreases as the models differ more, for example when transferring from SD1.4 to SDXL-Turbo compared to SD1.4 to SD2.1. For practical use, we therefore recommend running RATTPO directly on the target T2I model, especially since our experiments show that RATTPO works robustly across diverse T2I backbones (Table 6).

---

> ### Author Response · Authors · 2025-11-28
> **Kind Reminder Regarding Rebuttal**
>
> Dear Reviewer gfqR,
>
> We hope this message finds you well. We would like to kindly follow up regarding the rebuttal feedback for our submission. We truly appreciate the time and effort you have devoted to reviewing our paper, and we are happy to provide any clarifications if needed.
>
> We understand that this is a busy period and greatly value your consideration. Please let us know if any further information would be helpful from our side.
>
> With kind regards, \
> Authors of paper 11890

---

### Official Review · Reviewer_SnvR · 2025-11-01

**Soundness:** 2
**Presentation:** 3
**Contribution:** 2
**Rating:** 4
**Confidence:** 3

**Summary:**

This paper introduces RATTPO (Reward-Agnostic Test-Time Prompt Optimization), a novel framework for optimizing prompts in text-to-image (T2I) diffusion models without requiring reward-specific training or modifications. The method uses a dual-LLM approach:

An optimizer LLM iteratively refines prompts based on historical optimization trajectories.
A hint-generator LLM provides reward-aware feedback ("hints") derived from optimization history, replacing manual task descriptions.
RATTPO is training-free, gradient-free, and adaptable to diverse reward models (e.g., human preference, text-image alignment, multimodal LLM assessments). Experiments show it outperforms baselines in search efficiency (4.8× faster) and matches reward-specific methods with sufficient inference budget.

**Strengths:**

Extensive experiments across 8 reward setups, showing versatility and efficiency.

**Weaknesses:**

- Computational cost: Despite efficiency gains, RATTPO requires multiple image generations per iteration (line 7, Algorithm 1). Potential optimizations (e.g., caching) are unexplored.
- Prompt length constraints: The impact of initial prompt length on optimization is not analyzed.
- Novelty limited, because iteratively prompt optimization is trivial.

**Questions:**

See weakness.

---

> ### Author Response · Authors · 2025-11-21
> **Rebuttal Response [1/2]**
>
> We appreciate the reviewer's effort in providing feedback that helps us improve our work. We address the questions and weaknesses as below.
>
> ### **\[W1\] Computational cost and potential optimizations.**
>
> RATTPO is designed as a test-time optimization method, which leverages test-time compute in order to avoid the large training cost of learning-based prompt engineer (e.g., 4 GPU-Days to train PAG \[1\]). Within the class of test-time methods, RATTPO is also efficient, achieving on average a 4.8× speedup in wall-clock runtime compared to the reward-agnostic Paraphrase baseline to reach similar reward levels. **Taken together, this means RATTPO offers a favorable cost profile**, eliminating reward-specific training while remaining competitive in inference cost among search-based methods.
>
> We also note that the cost of RATTPO is configurable via the number of search rounds and candidates per round, based on the available budget. Furthermore, the multiple image generations within each round are naturally parallelizable. While further optimization is an interesting future direction, we believe RATTPO can be trivially combined with most optimization techniques for T2I models and LLMs. Regarding the potential optimization, we would appreciate further elaboration on the specific caching mechanism the reviewer envisions, as we are open to exploring its integration to enhance efficiency.
>
> \[1\] Yun et al. Learning to sample effective and diverse prompts for text-to-image generation. 2025\.
>
> ### **\[W2\] Analysis on the impact of initial prompt length on optimization**
>
> We conduct an additional analysis to identify how the initial prompt length affects the effectiveness of the prompt optimization. We group initial prompts into length quintiles and, for each bin, report the average absolute reward gain over the initial prompt. The experimental setup follows our ablation studies, and the plots are shown in the revised manuscript (Figure 24 in the Appendix J.1).
>
> **Table A: Average reward gain over the initial prompt, grouped by initial prompt length in the PR+Lexica setting. Plots are shown at Figure 24.**
>
> | Initial Prompt Length | 4-25 | 26-44 | 45-57 | 58-75 | 76-152 |
> | :---- | :---- | :---- | :---- | :---- | :---- |
> | Reward Gain\- Paraphrase | 1.036 | 0.734 | 0.643 | 0.538 | 0.522 |
> | Reward Gain\- RATTPO | 1.392 | 0.987 | 0.936 | 0.779 | 0.815 |
>
> **Table B: Average reward gain over the initial prompt, grouped by initial prompt length in the UniDet2D setting. Plots are shown at Figure 24.**
>
> | Initial Prompt Length | 17-25 | 26-28 | 29-30 | 31-33 | 34-37 |
> | :---- | :---- | :---- | :---- | :---- | :---- |
> | Reward Gain\- Paraphrase | 0.208 | 0.190 | 0.194 | 0.145 | 0.188 |
> | Reward Gain\- RATTPO | 0.269 | 0.326 | 0.300 | 0.227 | 0.305 |
>
> Across all length bins, RATTPO consistently achieves substantial reward improvements, and its gain is larger than that of the Paraphrase baseline in every bin, indicating that RATTPO effectively optimizes both short and long prompts. In the PR+Lexica setting, we further observe that shorter prompts benefit particularly strongly from optimization, which is intuitive since adding detailed scene, object, or style descriptions can substantially improve the aesthetics of generated images. In contrast, under the UniDet2D reward, no such correlation appears, likely because spatial composition scores do not necessarily benefit from longer or more descriptive prompts. These patterns illustrate that different reward models exhibit quite different prompt preferences. Despite this diversity, RATTPO adapts reliably to each underlying reward via the hint mechanism, yielding consistent, substantial gains across all bins. We have included the results and discussion in the revised manuscript, Appendix J.1.

---

> ### Author Response · Authors · 2025-11-21
> **Rebuttal Response [2/2]**
>
> ### **\[W3\] Novelty**
>
> We respectfully disagree with the assessment that iterative prompt optimization is “trivial”. While the high level idea of refining prompts based on feedback is simple to state, making such an iterative scheme work reliably in text to image generation under black-box, external rewards that are specified in test-time is not trivial in practice. While RATTPO successfully meets such requirements, naive strategies such as simply rephrasing prompts (Paraphrase baseline) or employing a single LLM loop without an adaptation mechanism (w/o hint, Table 2\) cannot, thus clearly underperform RATTPO.
>
> A key contribution of RATTPO is to address a significant but previously overlooked problem of reward agnostic T2I prompt engineering. Previous iterative schemes are designed for a single known reward or rely on a hand-written meta prompt that is tuned for a particular objective. In contrast, RATTPO is explicitly designed to adapt at test time to arbitrary reward models without retraining or manual redesign of the meta prompt.
>
> Furthermore, we propose a novel dual-LLM optimization loop to automate the role of the human written meta prompt. Prior multi LLM systems usually treat the second LLM as a static judge or scorer for a fixed task. In our case, the second LLM summarizes the trajectory into a concise textual description of how to increase the underlying reward, and the optimizer LLM then conditions on this learned strategy to produce the next prompt. This separation of roles between a trajectory summarizer that infers a reward aware strategy and an optimizer that uses this strategy for future updates is, to our knowledge, new in the context of T2I prompt engineering, and our ablations show that removing the hint-generator LLM clearly harms performance.
>
> In summary, we do not claim novelty in the mere existence of an iterative loop. Our contribution lies in how the loop is instantiated to solve the **reward agnostic** T2I prompt engineering problem, through a dual-LLM framework that automatically infers reward aware hints from trajectories and uses them to guide prompt updates. For a more detailed comparison between RATTPO and prior work, we also refer the reviewer to our rebuttal comment for gfqR \[W1\].

---

> ### Author Response · Authors · 2025-11-28
> **Kind Reminder Regarding Rebuttal**
>
> Dear Reviewer SnvR,
>
> We hope this message finds you well. We would like to kindly follow up regarding the rebuttal feedback for our submission. We truly appreciate the time and effort you have devoted to reviewing our paper, and we are happy to provide any clarifications if needed.
>
> We understand that this is a busy period and greatly value your consideration. Please let us know if any further information would be helpful from our side.
>
> With kind regards, \
> Authors of paper 11890

---

### Author Response · Authors · 2025-11-21
**Revision Summary**

We thank all reviewers for their thoughtful feedback. We have addressed each comment in the individual responses and updated the manuscript accordingly. To avoid confusion with figure and table numbering, all **newly added analyses are placed at the end of the paper (Appendix J-L, pp. 31-34)**. These updates will be later fully integrated into the final version of the manuscript.

Below is a concise summary of the main revisions:

### **Presentation**

- Added clarification on the search efficiency calculation (Appendix C)
- Revised one sentence for improved readability (L054).
- Added a visual illustration of the RATTPO workflow (Figure 26, Appendix L)

### **Additional Experiments and Analyses**

- Impact of initial prompt length on optimization performance (Figure 24, Appendix J.1)
- Experiments with GPT-family LLMs (Table 16, Appendix J.2).
- Comparison with a single-LLM variant (Table 17, Appendix J.3).
- Evaluation of T2I transferability of optimized prompts across models (Tables 18-19, Appendix J.4).
- Additional comparisons with two prior works (Figure 25 and Table 20, Appendix K).

We would be glad to clarify any remaining concerns and welcome further suggestions from the reviewers.

---

### Meta-Review · Area_Chair_D4u8 · 2026-01-05

**Summary:**

This paper works on test-time prompt optimization for text-to-image diffusion models. Authors proposed RATTPO applicable across various reward scenarios without modification. Experimental results verified the effectiveness of the proposed methods.

This paper got three 4 ratings and one 6 rating.

The strength of this paper given by reviewers are:
1. Extensive experiments. (Reviewer SnvR)
2. Experiments are promising. (Reviewer gfqR, oSLi)
3. algorithm is fairly simple and easy to implement. (Reviewer gfqR)
4. Paper is well written and easy to understand. (Reviewer gfqR)
5. well motivated. (Reviewer oSLi)
6. reward-agnostic nature of RATTPO. (Reviewer EEck)
7. method is training-free and gradient-free and exhibits superior generalization. (Reviewer EEck)
8. optimization process transparent and potentially human-interpretable. (Reviewer EEck)

The weakness of this paper given by reviewers are:
1. computational cost. (Reviewer SnvR, EEck)
2. Prompt length constraints. (Reviewer SnvR)
3. Novelty limited. (Reviewer SnvR, gfqR)
4. experiment results, while showing a lot of promise, do seem a bit selective and incomplete. (Reviewer gfqR)
5. need more experiments. (Reviewer gfqR)
6. paper is poorly written. (Reviewer oSLi)
7. lack a clear diagram illustrating the overall workflow. (Reviewer oSLi)
8. Lines 054-057 contain two sentences that redundantly express the same idea. (Reviewer oSLi)
9. Lack the ablation of using single prompting loop for both prompt generation and hint generation. (Reviewer EEck)

Questions:
1. How transferable are the prompts that are optimized for one text-to-image model to another one? (Reviewer gfqR)
2. Have you encountered the reward hacking problem in your optimization framework? (Reviewer EEck)
3. Why do you not consider integrating both optimizer and hint-generator in a single loop? (Reviewer EEck)

During rebuttal, Reviewer oSLi replied and mentioned they still have concerns on the quality of the paper writing and maintained score 4.

Given this concerns, novelty, computational cost concern, it is hard for AC to make a positive decision and has to reject this paper. Hope authors will find reviewers' feedback are useful for their future research.

**Reviewer Concerns:**

weakness 1. authors provided some answer. but if the model used many times, 4GPU-Days training cost is not that much?

weakness 2. authors provided more results to show gains for different initial prompt length.

weakness 3. authors provided explanations emphasize that reward agnostic is novel. This partial fixed reviewers concerns.

weakness 4. authors provided more results explained why some of the baselines are not included, and showed their method is better.

weakness 5. authors added more experiments and showed their method is better.

weakness 6, 7. authors revise the paper to make it better. but reviewer still has concerns on this.

weakness 8. authors removed the redundant sentences.

weakness 9. authors add ablation study for single prompting loop showing dual-LLM is better.

questions 1. results shows the transferable is not good.

questions 2. authors added some results. but it is less convincing given limited rewards used here. need to have better experimental design.

**Reviewer Scores:**

Reviewer SnvR might keep their score 4.

Reviewer gfqR might keep their score 4.

Reviewer oSLi mentioned they will keep their score 4.

Reviewer EEck might keep their score 6.

---

### Decision · Program_Chairs · 2026-01-26

Reject